# UNDERSTANDING DOMAIN RANDOMIZATION FOR SIM-TO-REAL TRANSFER

**Xiaoyu Chen & Jiachen Hu** *
Key Laboratory of Machine Perception, MOE,
School of Artificial Intelligence, Peking University
`{cxy30, NickH}@pku.edu.cn`

**Chi Jin**
Department of Electrical and Computer Engineering,
Princeton University
`chij@princeton.edu`

**Lihong Li**
Amazon
`llh@amazon.com`

**Liwei Wang**
Key Laboratory of Machine Perception, MOE,
School of Artificial Intelligence, Peking University
International Center for Machine Learning Research, Peking University
`wanglw@cis.pku.edu.cn`

## ABSTRACT

Reinforcement learning encounters many challenges when applied directly in the real world. Sim-to-real transfer is widely used to transfer the knowledge learned from simulation to the real world. Domain randomization—one of the most popular algorithms for sim-to-real transfer—has been demonstrated to be effective in various tasks in robotics and autonomous driving. Despite its empirical successes, theoretical understanding on why this simple algorithm works is limited. In this paper, we propose a theoretical framework for sim-to-real transfers, in which the simulator is modeled as a set of MDPs with tunable parameters (corresponding to unknown physical parameters such as friction). We provide sharp bounds on the sim-to-real gap—the difference between the value of policy returned by domain randomization and the value of an optimal policy for the real world. We prove that sim-to-real transfer can succeed under mild conditions without any real-world training samples. Our theory also highlights the importance of using memory (i.e., history-dependent policies) in domain randomization. Our proof is based on novel techniques that reduce the problem of bounding the sim-to-real gap to the problem of designing efficient learning algorithms for infinite-horizon MDPs, which we believe are of independent interest.

## 1 INTRODUCTION

Reinforcement Learning (RL) is concerned with sequential decision making, in which the agent interacts with the environment to maximize its cumulative rewards. This framework has achieved tremendous empirical successes in various fields such as Atari games, Go and StarCraft (Mnih et al., 2013; Silver et al., 2017; Vinyals et al., 2019). However, state-of-the-art algorithms often require a large amount of training samples to achieve such a good performance. While feasible in applications that have a good simulator such as the examples above, these methods are limited in applications where interactions with the real environment are costly and risky, such as healthcare and robotics.

One solution to this challenge is *sim-to-real transfer* (Floreano et al., 2008; Kober et al., 2013). The basic idea is to train an RL agent in a simulator that approximates the real world and then transfer the trained agent to the real environment. This paradigm has been widely applied, especially in robotics (Rusu et al., 2017; Peng et al., 2018; Chebotar et al., 2019) and autonomous driving (Pouyanfar et al., 2019; Niu et al., 2021). Sim-to-real transfer is appealing as it provides an essentially unlimited amount of data to the agent, and reduces the costs and risks in training.

However, sim-to-real transfer faces the fundamental challenge that the policy trained in the simulated environment may have degenerated performance in the real world due to the *sim-to-real gap*—the

---

*These two authors contributed equally.

mismatch between simulated and real environments. In addition to building higher-fidelity simulators to alleviate this gap, *domain randomization* is another popular method (Sadeghi & Levine, 2016; Tobin et al., 2017; Peng et al., 2018; OpenAI et al., 2018). Instead of training the agent in a single simulated environment, domain randomization randomizes the dynamics of the environment, thus exposes the agent to a diverse set of environments in the training phase. Policies learned entirely in the simulated environment with domain randomization can be directly transferred to the physical world with good performance (Sadeghi & Levine, 2016; Matas et al., 2018; OpenAI et al., 2018).

In this paper, we focus on understanding sim-to-real transfer and domain randomization from a theoretical perspective. The empirical successes raise the question: can we provide guarantees for the sub-optimality gap of the policy that is trained in a simulator with domain randomization and directly transferred to the physical world? To do so, we formulate the simulator as a set of MDPs with tunable latent variables, which corresponds to unknown parameters such as friction coefficient or wind velocity in the real physical world. We model the training process with domain randomization as finding an optimal history-dependent policy for a *latent MDP*, in which an MDP is randomly drawn from a set of MDPs in the simulator at the beginning of each episode.

Our contributions can be summarized as follows:

- We propose a novel formulation of sim-to-real transfer and establish the connection between domain randomization and the latent MDP model (Kwon et al., 2021). The latent MDP model illustrates the uniform sampling nature of domain randomization, and helps to analyze the sim-to-real gap for the policy obtained from domain randomization.

- We study the optimality of domain randomization in three different settings. Our results indicate that the sim-to-real gap of the policy trained in the simulation can be $o(H)$ when the randomized simulator class is finite or satisfies certain smoothness condition, where $H$ is the horizon of the real-world interaction. We also provide a lower bound showing that such benign conditions are necessary for efficient learning. Our theory highlights the importance of using memory (i.e., history-dependent policies) in domain randomization.

- To analyze the optimality of domain randomization, we propose a novel proof framework which reduces the problem of bounding the sim-to-real gap of domain randomization to the problem of designing efficient learning algorithms for infinite-horizon MDPs, which we believe are of independent interest.

- As a byproduct of our proof, we provide the first provably efficient model-based algorithm for learning infinite-horizon average-reward MDPs with general function approximation (Algorithm 4 in Appendix C.3). Our algorithm achieves a regret bound of $\tilde{O}(D\sqrt{d_e T})$, where $T$ is the total timesteps and $d_e$ is a complexity measure of a certain function class $\mathcal{F}$ that depends on the eluder dimension (Russo & Van Roy, 2013; Osband & Van Roy, 2014).

## 2 RELATED WORK

**Sim-to-Real and Domain Randomization** The basic idea of sim-to-real is to first train an RL agent in simulation, and then transfer it to the real environment. This idea has been widely applied to problems such as robotics (e.g., Ng et al., 2006; Bousmalis et al., 2018; Tan et al., 2018; OpenAI et al., 2018) and autonomous driving (e.g., Pouyanfar et al., 2019; Niu et al., 2021). To alleviate the influence of reality gap, previous works have proposed different methods to help with sim-to-real transfer, including progressive networks (Rusu et al., 2017), inverse dynamics models (Christiano et al., 2016) and Bayesian methods (Cutler & How, 2015; Pautrat et al., 2018). Domain randomization is an alternative approach to making the learned policy to be more adaptive to different environments (Sadeghi & Levine, 2016; Tobin et al., 2017; Peng et al., 2018; OpenAI et al., 2018), thus greatly reducing the number of real-world interactions.

There are also theoretical works related to sim-to-real transfer. Jiang (2018) uses the number of different state-action pairs as a measure of the gap between the simulator and the real environment. Under the assumption that the number of different pairs is constant, they prove the hardness of sim-to-real transfer and propose efficient adaptation algorithms with further conditions. Feng et al. (2019) prove that an approximate simulator model can effectively reduce the sample complexity in the real environment by eliminating sub-optimal actions from the policy search space. Zhong

et al. (2019) formulate a theoretical sim-to-real framework using the rich observation Markov decision processes (ROMDPs), and show that the transfer can result in a smaller real-world sample complexity. None of these results study benefits of domain randomization in sim-to-real transfer. Furthermore, all above works require real-world samples to fine-tune their policy during training, while our work and the domain randomization algorithm do not.

**POMDPs and Latent MDPs** Partially observable Markov decision processes (POMDPs) are a general framework for sequential decision-making problems when the state is not fully observable (Smallwood & Sondik, 1973; Kaelbling et al., 1998; Vlassis et al., 2012; Jin et al., 2020a; Xiong et al., 2021). Latent MDPs (Kwon et al., 2021), or LMDPs, are a special type of POMDPs, in which the real environment is randomly sampled from a set of MDPs at the beginning of each episode. This model has been widely investigated with different names such as hidden-model MDPs and multi-model MDPs. There are also results studying the planning problem in LMDPs, when the true parameters of the model is given (Chades et al., 2012; Buchholz & Scheftelowitsch, 2019; Steimle et al., 2021) . Kwon et al. (2021) consider the regret minimization problem for LMDPs, and provide efficient learning algorithms under different conditions. We remark that all works mentioned above focus on the problems of finding the optimal policies for POMDPs or latent MDPs, which is perpendicular to the central problem of this paper—bounding the performance gap of transferring the optimal policies of latent MDPs from simulation to the real environment.

**Infinite-horizon Average-Reward MDPs** Recent theoretical progress has produced many provably sample-efficient algorithms for RL in infinite-horizon average-reward setting. Nearly matching upper bounds and lower bounds are known for the tabular setting (Jaksch et al., 2010; Fruit et al., 2018; Zhang & Ji, 2019; Wei et al., 2020). Beyond the tabular case, Wei et al. (2021) propose efficient algorithms for infinite-horizon MDPs with linear function approximation. To the best of our knowledge, our result (Algorithm 4) is the first efficient algorithm with near-optimal regret for infinite-horizon average-reward MDPs with general function approximation.

## 3 PRELIMINARIES

### 3.1 EPISODIC MDPS

We consider episodic RL problems where each MDP is specified by $\mathcal{M} = (\mathcal{S}, \mathcal{A}, P, R, H, s_1)$. $\mathcal{S}$ and $\mathcal{A}$ are the state and the action space with cardinality $S$ and $A$ respectively. We assume that $S$ and $A$ are finite but can be extremely large. $P : \mathcal{S} \times \mathcal{A} \to \Delta(\mathcal{S})$ is the transition probability matrix so that $P(\cdot|s, a)$ gives the distribution over states if action $a$ is taken on state $s$, $R : \mathcal{S} \times \mathcal{A} \to [0, 1]$ is the reward function. $H$ is the number of steps in one episode.

For simplicity, we assume the agent always starts from the same state in each episode, and use $s_1$ to denote the initial state at step $h = 1$. It is straight-forward to extend our results to the case with random initialization. At step $h \in [H]$, the agent observes the current state $s_h \in \mathcal{S}$, takes action $a_h \in \mathcal{A}$, receives reward $R(s_h, a_h)$, and transits to state $s_{h+1}$ with probability $P(s_{h+1}|s_h, a_h)$. The episode ends when $s_{H+1}$ is reached.

We consider the history-dependent policy class $\Pi$, where $\pi \in \Pi$ is a collection of mappings from the history observations to the distributions over actions. Specifically, we use $traj_h = \{(s_1, a_1, s_2, a_2, \cdots, s_h) \mid s_i \in \mathcal{S}, a_i \in \mathcal{A}, i \in [h]\}$ to denote the set of all possible trajectories of history till step $h$. We define a policy $\pi \in \Pi$ to be a collection of $H$ policy functions $\{\pi_h : traj_h \to \Delta(\mathcal{A})\}_{h \in [H]}$. We define $V_{\mathcal{M},h}^{\pi} : \mathcal{S} \to \mathbb{R}$ to be the value function at step $h$ under policy $\pi$ on MDP $\mathcal{M}$, i.e., $V_{\mathcal{M},h}^{\pi}(s) = \mathbb{E}_{\mathcal{M},\pi}[\sum_{t=h}^{H} R(s_t, a_t) \mid s_h = s]$. Accordingly, we define $Q_{\mathcal{M},h}^{\pi} : \mathcal{S} \times \mathcal{A} \to R$ to be the Q-value function at step $h$: $Q_{\mathcal{M},h}^{\pi}(s, a) = \mathbb{E}_{\mathcal{M},\pi}[R(s_h, a_h) + \sum_{t=h+1}^{H} R(s_t, a_t) \mid s_h = s, a_h = a]$.

We use $\pi_{\mathcal{M}}^*$ to denote the optimal policy for a single MDP $\mathcal{M}$. It can be shown that there exists $\pi_{\mathcal{M}}^*$ such that the policy at step $h$ depends on only the state at step $h$ but not any other prior history. That is, $\pi_{\mathcal{M}}^*$ can be expressed as a collection of $H$ policy functions mapping from $\mathcal{S}$ to $\Delta(\mathcal{A})$. We use $V_{\mathcal{M},h}^*$ and $Q_{\mathcal{M},h}^*$ to denote the optimal value and Q-functions under the optimal policy $\pi_{\mathcal{M}}^*$ at step $h$.

## 3.2 Practical Implementation of Domain Randomization

In this subsection, we briefly introduce how domain randomization works in practical applications. Domain randomization is a popular technique for improving domain transfer (Tobin et al., 2017; Peng et al., 2018; Matas et al., 2018), which is often used for zero-shot transfer when the target domain is unknown or cannot be easily used for training. For example, by highly randomizing the rendering settings for their simulated training set, Sadeghi & Levine (2016) trained vision-based controllers for a quadrotor using only synthetically rendered scenes. OpenAI et al. (2018) studied the problem of dexterous in-hand manipulation. The training is performed entirely in a simulated environment in which they randomize the physical parameters of the system like friction coefficients and vision properties such as object's appearance.

To apply domain randomization in the simulation training, the first step before domain randomization is usually to build a simulator that is close to the real environment. The simulated model is further improved to match the physical system more closely through calibration. Though the simulation is still a rough approximation of the physical setup after these engineering efforts, these steps ensure that the randomized simulators generated by domain randomization can cover the real-world variability. During the training phase, many aspects of the simulated environment are randomized in each episode in order to help the agent learn a policy that generalizes to reality. The policy trained with domain randomization can be represented using recurrent neural network with memory such as LSTM (Yu et al., 2018; OpenAI et al., 2018; Doersch & Zisserman, 2019). Such a memory-augmented structure allows the policy to potentially identify the properties of the current environment and adapt its behavior accordingly. With sufficient data sampled using the simulator, the agent can find a near-optimal policy w.r.t. the average value function over a variety of simulation environments. This policy has shown its great adaptivity in many previous results, and can be directly applied to the physical world without any real-world fine-tuning (Sadeghi & Levine, 2016; Matas et al., 2018; OpenAI et al., 2018).

## 4 Formulation

In this section, we propose our theoretical formulation of sim-to-real and domain randomization. The corresponding models will be used to analyze the optimality of domain randomization in the next section, which can also serve as a starting point for future research on sim-to-real.

### 4.1 Sim-to-real Transfer

In this paper, we model the simulator as a set of MDPs with tunable latent parameters. We consider an MDP set $\mathcal{U}$ representing the simulator model with joint state space $\mathcal{S}$ and joint action space $\mathcal{A}$. Each MDP $\mathcal{M} = (\mathcal{S}, \mathcal{A}, P_{\mathcal{M}}, R, H, s_1)$ in $\mathcal{U}$ has its own transition dynamics $P_{\mathcal{M}}$, which corresponds to an MDP with certain choice of latent parameters. Our result can be easily extended to the case where the rewards are also influenced by the latent parameters. We assume that there exists an MDP $\mathcal{M}^* \in \mathcal{U}$ that represents the dynamics of the real environment.

We can now explain our general framework of sim-to-real. For simplicity, we assume that during the simulation phase (or training phase), we are given the entire set $\mathcal{U}$ that represents MDPs under different tunable latent parameter. Or equivalently, the learning agent is allowed to interact with any MDP $\mathcal{M} \in \mathcal{U}$ in arbitrary fashion, and sample arbitrary amount of trajectories. However, we do not know which MDP $\mathcal{M} \in \mathcal{U}$ represents the real environment. The objective of sim-to-real transfer is to find a policy $\pi$ purely based on $\mathcal{U}$, which performs well in the real environment. In particular, we measure the performance in terms of the *sim-to-real gap*, which is defined as the difference between the value of learned policy $\pi$ and the value of an optimal policy for the real world:

$$\text{Gap}(\pi) = V^*_{\mathcal{M}^*,1}(s_1) - V^\pi_{\mathcal{M}^*,1}(s_1). \tag{1}$$

We remark that in our framework, the policy $\pi$ is learned exclusively in simulation without the use of any real world samples. We study this framework because (1) our primary interests—domain randomization algorithm does not use any real-world samples for training; (2) we would like to focus on the problem of knowledge transfer from simulation to the real world. The more general learning paradigm that allows the fine-tuning of policy learned in simulation using real-world samples can

be viewed as a combination of sim-to-real transfer and standard on-policy reinforcement learning, which we left as an interesting topic for future research.

## 4.2 Domain Randomization and LMDPs

We first introduce Latent Markov decision processes (LMDPs) and then explain domain randomization in the viewpoint of LMDPs. A LMDP can be represented as $(\mathcal{U}, \nu)$, where $\mathcal{U}$ is a set of MDPs with joint state space $\mathcal{S}$ and joint action space $\mathcal{A}$, and $\nu$ is a distribution over $\mathcal{U}$. Each MDP $\mathcal{M} = (\mathcal{S}, \mathcal{A}, P_{\mathcal{M}}, R, H, s_1)$ in $\mathcal{U}$ has its own transition dynamics $P_{\mathcal{M}}$ that may differs from other MDPs. At the start of an episode, an MDP $\mathcal{M} \in \mathcal{U}$ is randomly chosen according to the distribution $\nu$. The agent does not know explicitly which MDP is sampled, but she is allowed to interact with this MDP $\mathcal{M}$ for one entire episode.

Domain randomization algorithm first specifies a distribution over tunable parameters, which equivalently gives a distribution $\nu$ over MDPs in simulator $\mathcal{U}$. This induces a LMDP with distribution $\nu$. The algorithm then samples trajectories from this LMDP, runs RL algorithms in order to find the near-optimal policy of this LMDP. We consider the ideal scenario that the domain randomization algorithm eventually find the globally optimal policy of this LMDP, which we formulate as domain randomization oracle as follows:

**Definition 1.** *(Domain Randomization Oracle) Let $\mathcal{U}$ be the set of MDPs generated by domain randomization and $\nu$ be the uniform distribution over $\mathcal{U}$. The domain randomization oracle returns an optimal history-dependent policy $\pi_{DR}^*$ of the LMDP $(\mathcal{U}, \nu)$:*

$$\pi_{DR}^* = \arg\max_{\pi \in \Pi} \mathbb{E}_{\mathcal{M} \sim \nu} V_{\mathcal{M},1}^{\pi}(s_1). \tag{2}$$

Since LMDP is a special case of POMDPs, its optimal policy $\pi_{\text{DR}}^*$ in general will depend on history. This is in sharp contrast with the optimal policy of a MDP, which is history-independent. We emphasize that both the memory-augmented policy and the randomization of the simulated environment are critical to the optimality guarantee of domain randomization. We also note that we don't restrict the learning algorithm used to find the policy $\pi_{\text{DR}}^*$, which can be either in a model-based or model-free style. Also, we don't explicitly define the behavior of $\pi_{\text{DR}}^*$. The only thing we know about $\pi_{\text{DR}}^*$ is that it satisfies the optimality condition defined in Equation 2. In this paper, we aim to bound the sim-to-real gap of $\pi_{\text{DR}}^*$, i.e., $\text{Gap}(\pi_{\text{DR}}^*, \mathcal{U})$ under different regimes.

## 5 Main Results

We are ready to present the sim-to-real gap of $\pi_{\text{DR}}^*$ in this section. We study the gap in three different settings under our sim-to-real framework: finite simulator class (the cardinality $|\mathcal{U}|$ is finite) with the separation condition (MDPs in $\mathcal{U}$ are distinct), finite simulator class without the separation condition, and infinite simulator class. During our analysis, we mainly study the long-horizon setting where $H$ is relatively large compared with other parameters. This is a challenging setting that has been widely-studied in recent years (Gupta et al., 2019; Mandlekar et al., 2020; Pirk et al., 2020). We show that the sim-to-real gap of $\pi_{\text{DR}}^*$ is only $O(\log^3(H))$ for the finite simulator class with the separation condition, and only $\tilde{O}(\sqrt{H})$ in the last two settings, matching the best possible lower bound in terms of $H$.

In our analysis, we assume that the MDPs in $\mathcal{U}$ are communicating MDPs with a bounded diameter.

**Assumption 1** (Communicating MDPs (Jaksch et al., 2010)). *The diameter of any MDP $\mathcal{M} \in \mathcal{U}$ is bounded by $D$. That is, consider the stochastic process defined by a stationary policy $\pi : \mathcal{S} \to \mathcal{A}$ on an MDP with initial state $s$. Let $T(s'|\mathcal{M}, \pi, s)$ denote the random variable for the first time step in which state $s'$ is reached in this process, then $\max_{s \neq s' \in \mathcal{S}} \min_{\pi : \mathcal{S} \to \mathcal{A}} \mathbb{E}\left[T\left(s' \mid \mathcal{M}, \pi, s\right)\right] \leq D$.*

This is a natural assumption widely used in the literature (Jaksch et al., 2010; Agrawal & Jia, 2017; Fruit et al., 2020). The communicating MDP model also covers many real-world tasks in robotics. For example, transferring the position or angle of a mechanical arm only costs constant time. Moreover, the diameter assumption is necessary under our framework.

**Proposition 1.** *Without Assumption 1, there exists a hard instance $\mathcal{U}$ so that $\text{Gap}(\pi_{DR}^*) = \Omega(H)$.*

We prove Proposition 1 in Appendix G.1. Note that the worst possible gap of any policy is $H$, so $\pi_{\mathrm{DR}}^*$ becomes ineffective without Assumption 1.

## 5.1 Finite Simulator Class With Separation Condition

As a starting point, we will show the sim-to-real gap when the MDP set $\mathcal{U}$ is a finite set with cardinality $M$. Intuitively, a desired property of $\pi_{\mathrm{DR}}^*$ is the ability to identify the environment the agent is exploring within a few steps. This is because $\pi_{\mathrm{DR}}^*$ is trained under uniform random environments, so we hope it can learn to tell the differences between environments. As long as $\pi_{\mathrm{DR}}^*$ has this property, the agent is able to identify the environment dynamics quickly, and behave optimally afterwards (note that the MDP set $\mathcal{U}$ is known to the agent).

Before presenting the general results, we first examine a simpler case where all MDPs in $\mathcal{U}$ are distinct. Concretely, we assume that any two MDPs in $\mathcal{U}$ are well-separated on at least one state-action pair. Note that this assumption is much weaker than the separation condition in Kwon et al. (2021), which assumes strongly separated condition for each state-action pair.

**Assumption 2** ($\delta$-separated MDP set). *For any $\mathcal{M}_1, \mathcal{M}_2 \in \mathcal{U}$, there exists a state-action pair $(s, a) \in \mathcal{S} \times \mathcal{A}$, such that the $L_1$ distance between the probability of next state of the different MDPs is at least $\delta$, i.e. $\|(P_{\mathcal{M}_1} - P_{\mathcal{M}_2})(\cdot \mid s, a)\|_1 \geq \delta$.*

The following theorem shows the sim-to-real gap of $\pi_{\mathrm{DR}}^*$ in $\delta$-separated MDP sets.

**Theorem 1.** *Under Assumption 1 and Assumption 2, for any $\mathcal{M} \in \mathcal{U}$, the sim-to-real gap of $\pi_{DR}^*$ is at most*

$$\mathrm{Gap}(\pi_{DR}^*) = O\left(\frac{DM^3 \log(MH) \log^2(SMH/\delta)}{\delta^4}\right). \tag{3}$$

The proof of Theorem 1 is deferred to Appendix D. Though the dependence on $M$ and $\delta$ may not be tight, our bound has only poly-logarithmic dependence on the horizon $H$.

The main difficulty to prove Theorem 1 is that we do not know what $\pi_{\mathrm{DR}}^*$ does exactly despite knowing a simple and clean strategy in the real-world interaction with minimum sim-to-real gap. That is, to firstly visit the state-action pairs that help the agent identify the environment quickly and then follow the optimal policy in the real MDP $\mathcal{M}^*$ after identifying $\mathcal{M}^*$. Therefore, we use a novel constructive argument in the proof. We construct a base policy that implements the idea mentioned above, and show that $\pi_{\mathrm{DR}}^*$ cannot be much worse than the base policy. The proof overview can be found in Section 6.

## 5.2 Finite Simulator Class Without Separation Condition

Now we generalize the setting and study the sim-to-real gap of $\pi_{\mathrm{DR}}^*$ when $\mathcal{U}$ is finite but not necessary a $\delta$-separated MDP set. Surprisingly, we show that $\pi_{\mathrm{DR}}^*$ can achieve $\tilde{O}(\sqrt{H})$ sim-to-real gap when $|\mathcal{U}| = M$.

**Theorem 2.** *Under Assumption 1, when the MDP set induced by domain randomization $\mathcal{U}$ is a finite set with cardinality $M$, the sim-to-real gap of $\pi_{DR}^*$ is upper bounded by*

$$\mathrm{Gap}(\pi_{DR}^*) = O\left(D\sqrt{M^3 H \log(MH)}\right). \tag{4}$$

Theorem 2 is proved in Appendix E. This theorem implies the importance of randomization and memory in the domain randomization algorithms (Sadeghi & Levine, 2016; Tobin et al., 2017; Peng et al., 2018; OpenAI et al., 2018). With both of them, we successfully reduce the worst possible gap of $\pi_{\mathrm{DR}}^*$ from the order of $H$ to the order of $\sqrt{H}$, so per step loss will be only $\tilde{O}(H^{-1/2})$. Without randomization, it is not possible to reduce the worst possible gap (i.e., the sim-to-real gap) because the policy is even not trained on all environments. Without memory, the policy is not able to implicitly "identify" the environments, so it cannot achieve sublinear loss in the worst case.

We also use a constructive argument to prove Theorem 2. However, it is more difficult to construct the base policy because we do not have any idea to minimize the gap without the well-separated condition (Assumption 2). Fortunately, we observe that the base policy is also a memory-based

policy, which basically can be viewed as an algorithm that seeks to minimize the sim-to-real gap in an unknown underlying MDP in $\mathcal{U}$. Therefore, we connect the sim-to-real gap of the base policy with the regret bound of the algorithms in *infinite-horizon average-reward MDPs* (Bartlett & Tewari, 2012; Fruit et al., 2018; Zhang & Ji, 2019). The proof overview is deferred to Section 6.

To illustrate the hardness of minimizing the worst case gap, we prove the following lower bound for $\mathrm{Gap}(\pi, \mathcal{U})$ to show that any policy must suffer a gap at least $\Omega(\sqrt{H})$.

**Theorem 3.** *Under Assumption 1, suppose $A \geq 10, SA \geq M \geq 100, D \geq 20 \log_A M, H \geq DM$, for any history dependent policy $\pi = \{\pi_h : traj_h \to \mathcal{A}\}_{h=1}^{H}$, there exists a set of $M$ MDPs $\mathcal{U} = \{\mathcal{M}_m\}_{m=1}^{M}$ and a choice of $\mathcal{M}^* \in \mathcal{U}$ such that $\mathrm{Gap}(\pi)$ is at least $\Omega(\sqrt{DMH})$.*

The proof of Theorem 3 follows the idea of the lower bound proof for tabular MDPs (Jaksch et al., 2010), which we defer to Appendix G.2. This lower bound implies that $\Omega(\sqrt{H})$ sim-to-real gap is unavoidable for the policy $\pi_{\mathrm{DR}}^*$ when directly transferred to the real environment.

### 5.3 Infinite Simulator Class

In real-world scenarios, the MDP class is very likely to be extensively large. For instance, many physical parameters such as surface friction coefficients and robot joint damping coefficients are sampled uniformly from a continuous interval in the Dexterous Hand Manipulation algorithms (OpenAI et al., 2018). In these cases, the induced MDP set $\mathcal{U}$ is large and even infinite. A natural question is whether we can extend our analysis to the infinite simulator class case, and provide a corresponding sim-to-real gap.

Intuitively, since the domain randomization approach returns the optimal policy in the average manner, the policy $\pi_{\mathrm{DR}}^*$ can perform bad in the real world $\mathcal{M}^*$ if most MDPs in the randomized set differ much with $\mathcal{M}^*$. In other words, $\mathcal{U}$ must be "smooth" near $\mathcal{M}^*$ for domain randomization to return a nontrivial policy. By "smoothness", we mean that there is a positive probability that the uniform distribution $\nu$ returns a MDP that is close to $\mathcal{M}^*$. This is because the probability that $\nu$ samples exactly $\mathcal{M}^*$ in a infinite simulator class is 0, so domain randomization cannot work at all if such smoothness does not hold.

Formally, we assume there is a distance measure $d(\mathcal{M}_1, \mathcal{M}_2)$ on $\mathcal{U}$ between two MDPs $\mathcal{M}_1$ and $\mathcal{M}_2$. Define the $\epsilon$-neighborhood $\mathcal{C}_{\mathcal{M}^*, \epsilon}$ of $\mathcal{M}^*$ as $\mathcal{C}_{\mathcal{M}^*, \epsilon} \overset{\text{def}}{=} \{\mathcal{M} \in \mathcal{U} : d(\mathcal{M}, \mathcal{M}^*) \leq \epsilon\}$. The smoothness condition is formally stated as follows:

**Assumption 3** (Smoothness near $\mathcal{M}^*$)**.** *There exists a positive real number $\epsilon_0$, and a Lipchitz constant $L$, such that for the policy $\pi_{DR}^*$, the value function of any two MDPs in $\mathcal{C}_{\mathcal{M}^*, \epsilon_0}$ is L-Lipchitz w.r.t the distance function d, i.e.*

$$\left| V_{\mathcal{M}_1, 1}^{\pi_{DR}^*}(s_1) - V_{\mathcal{M}_2, 1}^{\pi_{DR}^*}(s_1) \right| \leq L \cdot d(\mathcal{M}_1, \mathcal{M}_2), \forall \mathcal{M}_1, \mathcal{M}_2 \in \mathcal{C}_{\mathcal{M}^*, \epsilon_0}. \tag{5}$$

For example, we can set $d(\mathcal{M}_1, \mathcal{M}_2) = \mathbb{I}[\mathcal{M}_1 \neq \mathcal{M}_2]$ in the finite simulator class. For complicated simulator class, we need to ensure there exists some $d(\cdot, \cdot)$ that $L$ is not large.

With Assumption 3, it is possible to compute the sim-to-real gap of $\pi_{\mathrm{DR}}^*$. In the finite simulator class, we have shown that the gap depends on $M$ polynomially, which can be viewed as the complexity of $\mathcal{U}$. The question is, how do we measure the complexity of $\mathcal{U}$ when it is infinitely large?

Motivated by Ayoub et al. (2020), we consider the function class

$$\mathcal{F} = \{f_{\mathcal{M}}(s, a, \lambda) : \mathcal{S} \times \mathcal{A} \times \Lambda \to \mathbb{R} \text{ such that } f_{\mathcal{M}}(s, a, \lambda) = P_{\mathcal{M}}\lambda(s, a) \text{ for } \mathcal{M} \in \mathcal{U}, \lambda \in \Lambda\}, \tag{6}$$

where $\Lambda = \{\lambda_{\mathcal{M}}^*, \mathcal{M} \in \mathcal{U}\}$ is the optimal bias functions of $\mathcal{M} \in \mathcal{U}$ in the infinite-horizon average-reward setting (Bartlett & Tewari (2012); Fruit et al. (2018); Zhang & Ji (2019)). We note this function class is only used for analysis purposes to express our complexity measure; it does not affect the domain randomization algorithm. We use the the $\epsilon$-log-covering number and the $\epsilon$-eluder dimension of $\mathcal{F}$ to characterize the complexity of the simulator class $\mathcal{U}$. In the setting of linear combined models (Ayoub et al., 2020), the $\epsilon$-log-covering number and the $\epsilon$-eluder dimension are

$O\left(d\log(1/\epsilon)\right)$, where $d$ is the dimension of the linear representation in linear combined models. For readers not familiar with eluder dimension or infinite-horizon average-reward MDPs, please see Appendix A for preliminary explanations.

Here comes our bound of sim-to-real gap for the infinite simulator class setting, which is proved in Appendix F.

**Theorem 4.** *Under Assumption 1 and 3, the sim-to-real gap of the domain randomization policy $\pi_{DR}^*$ is at most for $0 \le \epsilon < \epsilon_0$*

$$\mathrm{Gap}(\pi_{DR}^*) = O\left(\frac{D\sqrt{d_e H \log\left(H \cdot \mathcal{N}(\mathcal{F}, 1/H)\right)}}{\nu\left(\mathcal{C}_{\mathcal{M}^*, \epsilon}\right)} + L\epsilon\right). \tag{7}$$

*Here $\nu(\mathcal{C}_{\mathcal{M}^*, \epsilon})$ is the probability of $\nu$ sampling a MDP in $\mathcal{C}_{\mathcal{M}^*, \epsilon}$, $d_e = dim_E(\mathcal{F}, 1/H)$ is the $1/H$-eluder dimension $\mathcal{F}$, and $\mathcal{N}(\mathcal{F}, 1/H)$ is the $1/H$-covering number of $\mathcal{F}$ w.r.t. $L_\infty$ norm.*

Theorem 4 is a generalization of Theorem 2, since we can reduce Theorem 4 to Theorem 2 by setting $d(\mathcal{M}_1, \mathcal{M}_2) = \mathbb{I}[\mathcal{M}_1 \ne \mathcal{M}_2]$ and $\epsilon = 0$, in which case $\nu(\mathcal{C}_{\mathcal{M}^*, \epsilon}) = 1/M$ and $d_e \le M$.

The proof overview can be found in Section 6. The main technique is still a reduction to the regret minimization problem in infinite-horizon average-reward setting. We construct a base policy and shows that the regret of it is only $\tilde{O}(\sqrt{H})$. A key point to note is that our construction of the base policy also solves an open problem of designing efficient algorithms that achieve $\tilde{O}(\sqrt{T})$ regret in the infinite-horizon average-reward setting with general function approximation. This base policy is of independent interests.

To complement our positive results, we also provide a negative result that even if the MDPs in $\mathcal{U}$ have nice low-rank properties (e.g., the linear low-rank property (Jin et al., 2020b; Zhou et al., 2020)), the policy $\pi_{\mathrm{DR}}^*$ returned by the domain randomization oracle can still have $\Omega(H)$ sim-to-real gap when the simulator class is large and the smoothness condition (Assumption 3) does not hold. This explains the necessity of our preconditions. Please refer to Proposition 2 in Appendix G.3 for details.

## 6 PROOF OVERVIEW

In this section, we will give a short overview of our novel proof techniques for the results shown in section 5. The main proof technique is based on reducing the problem of bounding the sim-to-real gap to the problem of constructing base policies. In the settings without separation conditions, we further connect the construction of the base policies to the design of efficient learning algorithms for the infinite-horizon average-reward settings.

### 6.1 REDUCING TO CONSTRUCTING BASE POLICIES

Intuitively, if there exists a base policy $\hat{\pi} \in \Pi$ with bounded sim-to-real gap, then the gap of $\pi_{\mathrm{DR}}^*$ will not be too large since $\pi_{\mathrm{DR}}^*$ defined in Eqn 2 is the policy with the maximum average value.

**Lemma 1.** *Suppose there exists a policy $\hat{\pi} \in \Pi$ such that the sim-to-real gap of $\hat{\pi}$ for any MDP $\mathcal{M} \in \mathcal{U}$ satisfies $V_{\mathcal{M},1}^*(s_1) - V_{\mathcal{M},1}^{\hat{\pi}}(s_1) \le C$, then we have $\mathrm{Gap}(\pi_{DR}^*) \le MC$ when $\mathcal{U}$ is a finite set with $|\mathcal{U}| = M$. Furthermore, when $\mathcal{U}$ is an infinite set satisfying the smoothness condition (assumption 3), we have for any $0 < \epsilon < \epsilon_0$, $\mathrm{Gap}(\pi_{DR}^*) \le C/\nu\left(\mathcal{C}_{\mathcal{M}^*, \epsilon}\right) + L\epsilon$.*

We defer the proof to Appendix B.1. Now with this reduction lemma, the remaining problem is defined as follows: Suppose the real MDP $\mathcal{M}^*$ belongs to the MDP set $\mathcal{U}$. We know the full information (transition matrix) of any MDP in the MDP set $\mathcal{U}$. How to design a history-dependent policy $\hat{\pi} \in \Pi$ with minimum sim-to-real gap $\max_{\mathcal{M} \in \mathcal{U}} \left(V_{\mathcal{M},1}^*(s_1) - V_{\mathcal{M},1}^{\hat{\pi}}(s_1)\right)$.

### 6.2 THE CONSTRUCTION OF THE BASE POLICIES

**With separation conditions** With the help of Lemma 1, we can bound the sim-to-real gap in the setting of finite simulator class with separation condition by constructing a history-dependent policy

$\hat{\pi}$. The formal definition of the policy $\hat{\pi}$ can be found in Appendix C.1. The idea of the construction is based on elimination: the policy $\hat{\pi}$ explicitly collects samples on the "informative" state-action pairs and eliminates the MDP that is less likely to be the real MDP from the candidate set. Once the agent identifies the real MDP representing the dynamics of the physical environment, it follows the optimal policy of the real MDP until the end of the interactions.

**Without separation conditions** The main challenge in this setting is that, we can no longer construct a policy $\hat{\pi}$ that "identify" the real MDP using the approaches as in the settings with separation conditions. In fact, we may not be able to even "identify" the real MDP since there can be MDPs in $\mathcal{U}$ that is very close to real MDP. Here, we use a different approach, which reduces the minimization of sim-to-real gap of $\hat{\pi}$ to the regret minimization problem in the infinite-horizon average-reward MDPs.

The infinite-horizon average-reward setting has been well-studied (e.g., Jaksch et al., 2010; Agrawal & Jia, 2017; Fruit et al., 2018; Wei et al., 2020). The main difference compared with the episodic setting is that the agent interacts with the environment for infinite steps. The gain of a policy is defined in the average manner. The value of a policy $\pi$ is defined as $\rho^\pi(s) = \mathbb{E}[\lim_{T \to \infty} \sum_{t=1}^{T} R(s_t, \pi(s_t))/T \mid s_1 = s]$. The optimal gain is defined as $\rho^*(s) \overset{\text{def}}{=} \max_{s \in \mathcal{S}} \max_\pi \rho^\pi(s)$, which is shown to be state-independent in Agrawal & Jia (2017), so we use $\rho^*$ for short. The regret in the infinite-horizon setting is defined as $\text{Reg}(T) = \mathbb{E}\left[T\rho^* - \sum_{t=1}^{T} R(s_t, a_t)\right]$, where the expectation is over the randomness of the trajectories. A more detailed explanation of infinite-horizon average-reward MDPs can be found in Appendix A.1.

For an MDP $\mathcal{M} \in \mathcal{U}$, we can view it as a finite-horizon MDP with horizon $H$; or we can view it as an infinite-horizon MDP. This is because Assumption 1 ensures that the agent can travel to any state from any state $s_H$ encountered at the $H$-th step (this may not be the case in the standard finite-horizon MDPs, since people often assume that the states at the $H$-th level are terminating state). The following lemma shows the connection between these two views.

**Lemma 2.** *For a MDP $\mathcal{M}$, let $\rho_{\mathcal{M}}^*$ and $V_{\mathcal{M},1}^*(s_1)$ to be the optimal expected gain in the infinite-horizon view and the optimal value function in the episodic view respectively. We have the following inequality: $H\rho_{\mathcal{M}}^* - D \leq V_{\mathcal{M},1}^*(s_1) \leq H\rho_{\mathcal{M}}^* + D$.*

This lemma indicates that, if we can design an algorithm (i.e. the base policy) $\hat{\pi}$ in the infinite-horizon setting with regret $\text{Reg}(H)$, then the sim-to-real gap of this algorithm in episodic setting satisfies $\text{Gap}(\hat{\pi}) = V_{\mathcal{M},1}^*(s_1) - V_{\mathcal{M},1}^{\hat{\pi}}(s_1) \leq \text{Reg}(H) + D$. This lemma connects the sim-to-real gap of $\hat{\pi}$ in finite-horizon setting to the regret in the infinite-horizon setting.

With the help of Lemma 1 and 2, the remaining problem is to design an efficient exploration algorithm for infinite-horizon average-reward MDPs with the knowledge that the real MDP $\mathcal{M}^*$ belongs to a known MDP set $\mathcal{U}$. Therefore, we propose two optimistic-exploration algorithms (Algorithm 3 and Algorithm 4) for the setting of finite simulator class and infinite simulator class respectively. The formal definition of the algorithms are deferred to Appendix C.2 and Appendix C.3. Note that our Algorithm 4 is the first efficient algorithm with $\tilde{O}(\sqrt{T})$ regret in the infinite-horizon average-reward MDPs with general function approximation, which is of independent interest for efficient online exploration in reinforcement learning.

## 7 CONCLUSION

In this paper, we study the optimality of policies learned from domain randomization in sim-to-real transfer without real-world samples. We propose a novel formulation of sim-to-real transfer and view domain randomization as an oracle that returns the optimal policy of an LMDP with uniform initialization distribution. Following this idea, we show that the policy $\pi_{\text{DR}}^*$ can suffer only $o(H)$ loss compared with the optimal value function of the real environment when the simulator class is finite or satisfies certain smoothness condition, thus this policy can perform well in the long-horizon cases. We hope our formulation and analysis can provide insight to design more efficient algorithms for sim-to-real transfer in the future.

## 8 ACKNOWLEDGMENTS

Liwei Wang was supported by National Key R&D Program of China (2018YFB1402600), Exploratory Research Project of Zhejiang Lab (No. 2022RC0AN02), BJNSF (L172037), Project 2020BD006 supported by PKUBaidu Fund.

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

## A  ADDITIONAL PRELIMINARIES

### A.1  INFINITE-HORIZON AVERAGE-REWARD MDPS

The infinite-horizon average-reward setting has been well-explored in the recent few years (e.g. Jaksch et al. (2010); Agrawal & Jia (2017); Fruit et al. (2018); Wei et al. (2020)). The main difference compared with the episodic setting is that the agent interacts with the environment for infinite steps instead of restarting every $H$ steps. The gain of a policy is defined in the average manner.

**Definition 2.** *(Definition 4 in Agrawal & Jia (2017)) The gain $\rho^\pi(s)$ of a stationary policy $\pi$ from starting state $s_1 = s$ is defined as:*

$$\rho^\pi(s) = \mathbb{E}\left[\lim_{T\to\infty}\frac{1}{T}\sum_{t=1}^{T} R(s_t, \pi(s_t)) \mid s_1 = s\right] \tag{8}$$

In this setting, a common assumption is that the MDP is communicating (Assumption 1). Under this assumption, we have the following lemma.

**Lemma 3.** *(Agrawal & Jia, 2017, Lemma 2.1) For a communicating MDP $\mathcal{M}$ with diameter $D$: (a) The optimal gain $\rho^*$ is state-independent and is achieved by a deterministic stationary policy $\pi_{DR}^*$; that is, there exists a deterministic policy $\pi^*$ such that*

$$\rho^* := \max_{s'\in\mathcal{S}}\max_\pi \rho^\pi(s') = \rho^{\pi^*}(s), \forall s \in \mathcal{S} \tag{9}$$

*(b) The optimal gain $\rho^*$ satisfies the following equation:*

$$\rho^* = \min_{\lambda\in\mathbb{R}^S}\max_{s,a}\left[R(s,a) + P\lambda(s,a) - \lambda(s)\right] = \max_a\left[R(s,a) + P\lambda^*(s,a) - \lambda^*(s)\right], \forall s \tag{10}$$

*where $P\lambda(s,a) = \sum_{s'} P(s'|s,a)\lambda(s')$, and $\lambda^*$ is the bias vector of the optimal policy $\pi_{DR}^*$ satisfying*

$$0 \le \lambda^*(s) \le D. \tag{11}$$

The regret minimization problem has been widely studied in this setting, with regret to be defined as $Reg(T) = \mathbb{E}\left[T\rho^* - \sum_{t=1}^T R(s_t, a_t)\right]$, where the expectation is over the randomness of the trajectories. For example, Jaksch et al. (2010) proposed an efficient algorithm called UCRL2, which achieves regret upper bound $\tilde{O}(DS\sqrt{AT})$. For notation convenience, we use $PV(s,a)$ or $P\lambda(s,a)$ as a shorthand of $\sum_{s'\in\mathcal{S}} P(s'|s,a)V(s')$ or $\sum_{s'\in\mathcal{S}} P(s'|s,a)\lambda(s')$.

## A.2  Eluder Dimension

Proposed by Russo & Van Roy (2013), eluder dimension has become a widely-used concept to characterize the complexity of different function classes in bandits and RL (Wang et al., 2020; Ayoub et al., 2020; Jin et al., 2021; Kong et al., 2021). In this work, we define eluder dimension to characterize the complexity of the function $\mathcal{F}$:

$$\mathcal{F} = \{f_{\mathcal{M}}(s, a, \lambda) : \mathcal{S} \times \mathcal{A} \times \Lambda \to \mathbb{R} \text{ such that } f_{\mathcal{M}}(s, a, \lambda) = P_{\mathcal{M}}\lambda(s, a) \text{ for } \mathcal{M} \in \mathcal{U}, \lambda \in \Lambda\},$$
(12)

where $\Lambda = \{\lambda^*_{\mathcal{M}}, \mathcal{M} \in \mathcal{U}\}$ is the optimal bias functions of $\mathcal{M} \in \mathcal{U}$ in the infinite-horizon average-reward setting (Bartlett & Tewari (2012); Fruit et al. (2018); Zhang & Ji (2019)).

**Definition 3.** *(Eluder dimension). Let $\epsilon \geq 0$ and $\mathcal{Z} = \{(s_i, a_i, \lambda_i)\}_{i=1}^n \subset \mathcal{S} \times \mathcal{A} \times \Lambda$ be a sequence of history samples.*

- *A history sample $(s, a, \lambda) \in \mathcal{S} \times \mathcal{A} \times \Lambda$ is $\epsilon$-dependent on $\mathcal{Z}$ with respect to $\mathcal{F}$ if any $f, f' \in \mathcal{F}$ satisfying $\|f - f'\|_{\mathcal{Z}} \leq \epsilon$ also satisfies $\|f(s, a) - f'(s, a)\| \leq \epsilon$. Here $\|f - f'\|_{\mathcal{Z}}$ is a shorthand of $\sqrt{\sum_{(s,a,\lambda) \in \mathcal{Z}}(f - f')^2(s, a, \lambda)}$.*

- *An $(s, a, \lambda)$ is $\epsilon$-independent of $\mathcal{Z}$ with respect to $\mathcal{F}$ if $(s, a, \lambda)$ is not $\epsilon$-dependent on $\mathcal{Z}$.*

- *The $\epsilon$-eluder dimension of a function class $\mathcal{F}$ is the length of the longest sequence of elements in $\mathcal{S} \times \mathcal{A} \times \Lambda$ such that, for some $\epsilon' \geq \epsilon$, every element is $\epsilon'$-independent of its predecessors.*

## B  Omitted Proof in Section 6

### B.1  Proof of Lemma 1

*Proof.* We firstly study the case where $\mathcal{U}$ is a finite set with $|\mathcal{U}| = M$. For $\hat{\pi}$, we have

$$\frac{1}{M} \sum_{\mathcal{M} \in \mathcal{U}} \left(V^*_{\mathcal{M},1}(s_1) - V^{\hat{\pi}}_{\mathcal{M},1}(s_1)\right) \leq C.$$
(13)

By the optimality of $\pi^*_{\text{DR}}$, we know that

$$\frac{1}{M} \sum_{\mathcal{M} \in \mathcal{U}} V^{\pi^*_{\text{DR}}}_{\mathcal{M},1}(s_1) \geq \frac{1}{M} \sum_{\mathcal{M} \in \mathcal{U}} V^{\hat{\pi}}_{\mathcal{M},1}(s_1).$$
(14)

Therefore,

$$\frac{1}{M} \sum_{\mathcal{M} \in \mathcal{U}} \left(V^*_{\mathcal{M},1}(s_1) - V^{\pi^*_{\text{DR}}}_{\mathcal{M},1}(s_1)\right) \leq C.$$
(15)

Since the gap $V^*_{\mathcal{M},1}(s_1) - V^{\pi^*_{\text{DR}}}_{\mathcal{M},1}(s_1) \geq 0$ for any $i \in [M]$, we have $\frac{1}{M}\left(V^*_{M^*,1}(s_1) - V^{\pi^*_{\text{DR}}}_{M^*,1}(s_1)\right) \leq C$. That is,

$$\left(V^*_{M^*,1}(s_1) - V^{\pi^*_{\text{DR}}}_{M^*,1}(s_1)\right) \leq MC.$$
(16)

For the case where $\mathcal{U}$ is an infinite set satisfying Assumption 3, by the optimality of $\pi^*_{\text{DR}}$, we have

$$\mathbb{E}_{\mathcal{M} \sim \nu}\left[V^{\pi^*_{\text{DR}}}_{\mathcal{M},1}(s_1)\right] \geq \mathbb{E}_{\mathcal{M} \sim \nu}\left[V^{\hat{\pi}}_{\mathcal{M},1}(s_1)\right].$$
(17)

Therefore,

$$\mathbb{E}_{\mathcal{M} \sim \nu(\mathcal{C}_{\mathcal{M}^*,\epsilon})}\left[V^*_{M^*,1}(s_1) - V^{\pi^*_{\text{DR}}}_{\mathcal{M},1}(s_1)\right] \leq \mathbb{E}_{\mathcal{M} \sim \nu}\left[V^*_{M^*,1}(s_1) - V^{\pi^*_{\text{DR}}}_{\mathcal{M},1}(s_1)\right] \leq \mathbb{E}_{\mathcal{M} \sim \nu}\left[V^*_{M^*,1}(s_1) - V^{\hat{\pi}}_{\mathcal{M},1}(s_1)\right].$$
(18)

By Assumption 3, for any $\mathcal{M} \in \mathcal{C}(\mathcal{M}^*, \epsilon)$, we have

$$\left| V_{\mathcal{M}^*,1}^{\pi_{\mathrm{DR}}^*}(s_1) - V_{\mathcal{M},1}^{\pi_{\mathrm{DR}}^*}(s_1) \right| \leq L\epsilon. \tag{19}$$

Therefore, we have

$$\nu\left(\mathcal{C}_{\mathcal{M}^*,\epsilon}\right)\left(V_{\mathcal{M}^*,1}^*(s_1) - V_{\mathcal{M}^*,1}^{\pi_{\mathrm{DR}}^*}(s_1) - L\epsilon\right) \leq \mathbb{E}_{\mathcal{M}\sim\nu(\mathcal{C}_{\mathcal{M}^*,\epsilon})}\left[V_{\mathcal{M}^*,1}^*(s_1) - V_{\mathcal{M},1}^{\pi_{\mathrm{DR}}^*}(s_1)\right] \tag{20}$$

Combining Inq 18 and Inq 20, we have

$$\nu\left(\mathcal{C}_{\mathcal{M}^*,\epsilon}\right)\left(V_{\mathcal{M}^*,1}^*(s_1) - V_{\mathcal{M}^*,1}^{\pi_{\mathrm{DR}}^*}(s_1) - L\epsilon\right) \leq C, \tag{21}$$

The lemma can be proved by reordering the above inequality. □

### B.2 PROOF OF LEMMA 2

*Proof.* For MDP $\mathcal{M}$, denote $\pi_{in}^*$ as the optimal policy in the infinite-horizon setting and $\{\pi_{ep,h}^*\}_{h=1}^H$ as the optimal policy in the episodic setting. By the optimality of $\pi_{ep}^*$, we have $V_{\mathcal{M},1}^*(s_1) = V_{\mathcal{M},1}^{\pi_{ep}^*}(s_1) \geq V_{\mathcal{M},1}^{\pi_{in}^*}(s_1)$.

By the Bellman equation in the infinite-horizon setting, we know that

$$\lambda_{\mathcal{M}}^*(s) + \rho_{\mathcal{M}}^* = R(s, \pi_{in}^*(s)) + P_{\mathcal{M}}\lambda_{\mathcal{M}}^*(s, \pi_{in}^*(s)), \forall s \in \mathcal{S} \tag{22}$$

For notation simplicity, we use $d_h(s_1, \pi)$ to denote the state distribution at step $h$ after starting from state $s_1$ at step 1 following policy $\pi$. From the above equation, we have

$$\lambda_{\mathcal{M}}^*(s_1) + H\rho_{\mathcal{M}}^* = \sum_{h=1}^H \mathbb{E}_{s_h \sim d_h(s_1, \pi_{in}^*)} R(s_h, , \pi_{in}^*(s_h)) + \mathbb{E}_{s_{H+1} \sim d_{H+1}(s_1, \pi_{in}^*)} \lambda_{\mathcal{M}}^*(s_{H+1}). \tag{23}$$

That is,

$$|\sum_{h=1}^H \mathbb{E}_{s_h \sim d_h(s_1, \pi_{in}^*)} R(s_h, , \pi_{in}^*(s_h)) - H\rho_{\mathcal{M}}^*| = |\lambda_{\mathcal{M}}^*(s_1) - \mathbb{E}_{s_{H+1} \sim d_{H+1}(s_1, \pi_{in}^*)} \lambda_{\mathcal{M}}^*(s_{H+1})| \leq D,$$
$$\tag{24}$$

where $\sum_{h=1}^H \mathbb{E}_{s_h \sim d_h(s_1, \pi_{in}^*)} R(s_h, \pi_{in}^*(s_h)) = V_{\mathcal{M},1}^{\pi_{in}^*}(s_1)$. Therefore, we have $H\rho_{\mathcal{M}}^* - D \leq V_{\mathcal{M},1}^{\pi_{in}^*}(s_1) \leq V_{\mathcal{M},1}^*(s_1)$.

For the second inequality, by the Bellman equation in the infinite-horizon setting, we have

$$\lambda_{\mathcal{M}}^*(s_1) + H\rho_{\mathcal{M}}^* \geq \sum_{h=1}^H \mathbb{E}_{s_h \sim d_h(s_1, \pi_{ep}^*)} R(s_h, , \pi_{ep,h}^*(s_h)) + \mathbb{E}_{s_{H+1} \sim d_{H+1}(s_1, \pi_{ep}^*)} \lambda_{\mathcal{M}}^*(s_{H+1}).$$
$$\tag{25}$$

That is,

$$\sum_{h=1}^H \mathbb{E}_{s_h \sim d_h(s_1, \pi_{ep}^*)} R(s_h, , \pi_{ep,h}^*(s_h)) - H\rho_{\mathcal{M}}^* \leq \lambda_{\mathcal{M}}^*(s_1) - \mathbb{E}_{s_{H+1} \sim d_{H+1}(s_1, \pi_{ep}^*)} \lambda_{\mathcal{M}}^*(s_{H+1}) \leq D,$$
$$\tag{26}$$

where $\sum_{h=1}^H \mathbb{E}_{s_h \sim d_h(s_1, \pi_{ep}^*)} R(s_h, , \pi_{ep,h}^*(s_h)) = V_{\mathcal{M},1}^*(s_1)$. □

## C THE CONSTRUCTION OF LEARNING ALGORITHMS

### C.1 FINITE SIMULATOR CLASS WITH SEPARATION CONDITION

In this subsection, we explicitly define the base policy $\hat{\pi}$ with sim-to-real gap guarantee under the separation condition. Note that a history-dependent policy for LMDPs can also be regarded as an

---

**Algorithm 1** Optimistic Exploration Under Separation Condition

---

1: Initialize: the MDP set $\mathcal{D} = \mathcal{U}$, $n_0 = \frac{c_0 \log^2(SMH) \log(MH)}{\delta^4}$ for a constant $c_0$

2: ▷ *Stage 1: Explore and find the real MDP $\mathcal{M}^*$*

3: **while** $|\mathcal{D}| \geq 1$ **do**

4:      Randomly select two MDPs $\mathcal{M}_1$ and $\mathcal{M}_2$ from the MDP set $\mathcal{D}$

5:      Choose $(s_0, a_0) = \arg\max_{(s,a) \in \mathcal{S} \times \mathcal{A}} \|(P_{\mathcal{M}_1} - P_{\mathcal{M}_2})(\cdot \mid s, a)\|_1$

6:      Call Subroutine 2 with parameter $(s_0, a_0)$ and $n_0$ to collect history samples $\mathcal{H}_{\mathcal{M}_1, \mathcal{M}_2}$

7:      **if** $\exists s' \in \mathcal{H}_{\mathcal{M}_1, \mathcal{M}_2}, P_{\mathcal{M}_2}(s'|s_0, a_0) = 0$ or $\prod_{s' \in \mathcal{H}_{\mathcal{M}_1, \mathcal{M}_2}} \frac{P_{\mathcal{M}_1}(s'|s_0, a_0)}{P_{\mathcal{M}_2}(s'|s_0, a_0)} \geq 1$ **then**

8:          Eliminate $\mathcal{M}_2$ from the MDP set $\mathcal{D}$

9:      **else**

10:          Eliminate $\mathcal{M}_1$ from the MDP set $\mathcal{D}$

11:      **end if**

12: **end while**

13: ▷ *Stage 2: Run the optimal policy of $\mathcal{M}^*$*

14: Denote $\hat{\mathcal{M}}$ as the remaining MDP in the MDP set $\mathcal{D}$

15: Run the optimal policy of $\hat{\mathcal{M}}$ for the remaining steps

---

**Algorithm 2** Subroutine: collecting data for $(s_0, a_0)$

---

     Input: informative state-action pair $(s_0, a_0)$, required visitation count $n_0$

     Initialize: counter $N(s_0, a_0) = 0$, history data $\mathcal{H} = \emptyset$

     Denote $\pi_{\mathcal{M}}(s, s')$ as the policy with the minimum expected travelling time $\mathbb{E}[T(s' \mid \mathcal{M}, \pi, s)]$ for MDP $\mathcal{M}$ (Defined in Assumption 1)

     **while** $N(s_0, a_0) \leq n_0$ **do**

5:      **for** $i = 1, \cdots, M$ **do**

         Denote the current state as $s_{init}$

         Run the policy $\pi_{\mathcal{M}_i}(s_{init}, s_0)$ for $2D$ steps, breaking the loop immediately once the agent enters state $s_0$

     **end for**

     **if** the agent enters state $s_0$ **then**

10:          Execute $a_0$, enter the next state $s'$.

         counter $N(s_0, a_0) = N(s_0, a_0) + 1$, $\mathcal{H} = \mathcal{H} \bigcup \{s'\}$

     **end if**

     **end while**

     Output: history data $\mathcal{H}$

---

algorithm for finite-horizon MDPs. By deriving an upper bound of the sim-to-real gap for $\hat{\pi}$, we can upper bound $\mathrm{Gap}(\pi^*_{\mathrm{DR}}, \mathcal{U})$ with Lemma 1.

The policy $\hat{\pi}$ is formally defined in Algorithm 1. There are two stages in Algorithm 1. In the first stage, the agent's goal is to quickly explore the environment and find the real MDP $\mathcal{M}^*$ from the MDP set $\mathcal{U}$. This stage contains at most $M-1$ parts. In each part, the agent randomly selects two MDPs $\mathcal{M}_1$ and $\mathcal{M}_2$ from the remaining MDP set $\mathcal{D}$. Since the agent knows the transition dynamics of $\mathcal{M}_1$ and $\mathcal{M}_2$, it can find the most informative state-action pair $(s_0, a_0)$ with maximum total-variation difference between $P_{\mathcal{M}_1}(\cdot|s_0, a_0)$ and $P_{\mathcal{M}_2}(\cdot|s_0, a_0)$. The algorithm calls Subroutine 2 to collect enough samples from $(s_0, a_0)$ pairs, and then eliminates the MDP with less likelihood. At the end of the first stage, the MDP set $\mathcal{D}$ is ensured to contain only one MDP $\mathcal{M}^*$ with high probability. Therefore, the agent can directly execute the optimal policy for the real MDP till step $H+1$ in the second stage.

Subroutine 2 is designed to collect enough samples from the given state-action pair $(s_0, a_0)$. The basic idea in Subroutine 2 is to quickly enter the state $s_0$ and take action $a_0$, until the visitation counter $N(s_0, a_0) = n_0$. Denote $\pi_{\mathcal{M}}(s, s')$ as the policy with the minimum expected travelling time $\mathbb{E}[T(s' \mid \mathcal{M}, \pi, s)]$ for MDP $\mathcal{M}$. Suppose the agent is currently at state $s_{init}$ and runs the policy $\pi_{\mathcal{M}^*}(s_{init}, s_0)$ in the following steps. By Assumption 1 and Markov's inequality, the agent will enter state $s_0$ in $2D$ steps with probability at least $1/2$. Therefore, in Subroutine 2, we runs the policy $\pi_{\mathcal{M}_i}(s_{init}, s_0)$ for $2D$ steps for $i \in [M]$ alternatively. This ensures that the agent can enter state $s_0$ in $2MD$ steps with probability at least $1/2$.

Theorem 5 states an upper bound of the sim-to-real gap for Algorithm 1, which is proved in Appendix D.1.

**Theorem 5.** *Suppose we use $\hat{\pi}$ to denote the history-dependent policy represented by Algorithm 1. Under Assumption 1 and Assumption 2, for any $\mathcal{M} \in \mathcal{U}$, the sim-to-real gap of Algorithm 1 is at most*

$$V^*_{\mathcal{M},1}(s_1) - V^{\hat{\pi}}_{\mathcal{M},1}(s_1) \leq O\left(\frac{DM^2 \log(MH) \log^2(SMH/\delta)}{\delta^4}\right). \tag{27}$$

### C.2 Finite Simulator Class without Separation Condition

In this subsection, we propose an efficient algorithm in the infinite-horizon average-reward setting for finite simulator class. Our algorithm is described in Algorithm 3. In episode $k$, the agent executes the optimal policy of the optimistic MDP $\mathcal{M}^k$ with the maximum expected gain $\rho^*_{\mathcal{M}^k}$. Once the agent collects enough data and realizes that the current MDP $\mathcal{M}^k$ is not $\mathcal{M}^*$ that represents the dynamics of the real environment, the agent eliminates $\mathcal{M}^k$ from the MDP set.

---

**Algorithm 3** Optimistic Exploration

---

1: Initialize: the MDP set $\mathcal{U}_1 = \mathcal{U}$, the episode counter $k = 1$, $h_0 = 1$
2: Calculate $\mathcal{M}^1 = \arg\max_{\mathcal{M} \in \mathcal{U}_1} \rho^*_{\mathcal{M}}$
3: **for** step $h = 1, \cdots, H$ **do**
4:     Take action $a_h = \pi^*_{\mathcal{M}^k}(s_h)$, obtain the reward $R(s_h, a_h)$, and observe the next state $s_{h+1}$
5:     **if** $\left|\sum_{t=h_0}^h \left(P_{\mathcal{M}^k}\lambda^*_{\mathcal{M}^k}(s_t, a_t) - \lambda^*_{\mathcal{M}^k}(s_{t+1})\right)\right| > D\sqrt{2(h-h_0)\log(2HM)}$ **then**
6:         Eliminate $\mathcal{M}^k$ from the MDP set $\mathcal{U}_k$, denote the remaining set as $\mathcal{U}_{k+1}$
7:         Calculate $\mathcal{M}^{k+1} = \arg\max_{\mathcal{M} \in \mathcal{U}_{k+1}} \rho^*_{\mathcal{M}}$
8:         Set $h_0 = h + 1$, and $k = k + 1$.
9:     **end if**
10: **end for**

---

To indicate the basic idea of the elimination condition defined in Line 5 of Algorithm 3, we briefly explain our regret analysis of Algorithm 3. Suppose the MDP $\mathcal{M}^k$ selected in episode $k$ satisfies the

optimistic condition $\rho^*_{\mathcal{M}^k} \geq \rho^*_{\mathcal{M}^*}$, then the regret in $H$ steps can be bounded as:

$$H\rho^*_{\mathcal{M}^*} - \sum_{h=1}^{H} R(s_h, a_h) \tag{28}$$

$$\leq \sum_{k=1}^{K} \sum_{h=\tau(k)}^{\tau(k+1)-1} (\rho^*_{\mathcal{M}^k} - R(s_h, a_h)) \tag{29}$$

$$= \sum_{k=1}^{K} \sum_{h=\tau(k)}^{\tau(k+1)-1} (P_{\mathcal{M}^k}\lambda^*_{\mathcal{M}^k}(s_h, a_h) - \lambda^*_{\mathcal{M}^k}(s_h)) \tag{30}$$

$$= \sum_{k=1}^{K} \sum_{h=\tau(k)}^{\tau(k+1)-1} (P_{\mathcal{M}^k}\lambda^*_{\mathcal{M}^k}(s_h, a_h) - \lambda^*_{\mathcal{M}^k}(s_{h+1})) + \sum_{k=1}^{K} \left(\lambda^*_{\mathcal{M}^k}(s_{\tau(k+1)}) - \lambda^*_{\mathcal{M}^k}(s_{\tau(k)})\right) \tag{31}$$

$$\leq \sum_{k=1}^{K} \sum_{h=\tau(k)}^{\tau(k+1)-1} (P_{\mathcal{M}^k}\lambda^*_{\mathcal{M}^k}(s_h, a_h) - \lambda^*_{\mathcal{M}^k}(s_{h+1})) + KD. \tag{32}$$

Here we use $K$ to denote the total number of episodes, and we use $\tau(k)$ to denote the first step of episode $k$. The first inequality is due to the optimistic condition $\rho^*_{\mathcal{M}^k} \geq \rho^*_{\mathcal{M}^*}$. The first equality is due to the Bellman equation in the finite-horizon setting (Eqn 10). The last inequality is due to $0 \leq \lambda^*_{\mathcal{M}}(s) \leq D$. From the above inequality, we know that the regret in episode $k$ depends on the summation $\sum_{h=\tau(k)}^{\tau(k+1)-1} \left(P_{\mathcal{M}^k}\lambda^*_{\mathcal{M}^k}(s_h, a_h) - \lambda^*_{\mathcal{M}^k}(s_{h+1})\right)$. If this term is relatively small, we can continue following the policy $\pi^*_{\mathcal{M}^k}$ with little loss. Since $\lambda^*_{\mathcal{M}^k}(s_{h+1})$ is an empirical sample of $P_{\mathcal{M}^*}\lambda^*_{\mathcal{M}^k}(s_h, a_h)$, we can guarantee that $\mathcal{M}^k$ is not $\mathcal{M}^*$ with high probability if this term is relatively large.

Based on the above discussion, we can upper bound the regret of Algorithm 3. We defer the proof of Theorem 6 to Appendix E.1.

**Theorem 6.** *Under Assumption 1, the regret of Algorithm 3 is upper bounded by*

$$Reg(H) \leq O\left(D\sqrt{MH\log(MH)}\right). \tag{33}$$

### C.3 INFINITE SIMULATOR CLASS

In this subsection, we propose a provably efficient model-based algorithm solving infinite-horizon average-reward MDPs with general function approximation. To the best of our knowledge, our result is the first efficient algorithm with near-optimal regret for infinite-horizon average-reward MDPs with general function approximation.

Considering the model class $\mathcal{U}$ which covers the true MDP $\mathcal{M}^*$, i.e. $\mathcal{M}^* \in \mathcal{U}$, we define the function space $\Lambda = \{\lambda^*_{\mathcal{M}}, \mathcal{M} \in \mathcal{U}\}$, and space $\mathcal{X} = \mathcal{S} \times \mathcal{A} \times \Lambda$. We define the function space

$$\mathcal{F} = \{f_{\mathcal{M}}(s, a, \lambda) : \mathcal{X} \to \mathbb{R} \text{ such that } f_{\mathcal{M}}(s, a, \lambda) = P_{\mathcal{M}}\lambda(s, a) \text{ for } \mathcal{M} \in \mathcal{U}, \lambda \in \Lambda\}. \tag{34}$$

Our algorithm, which is described in Algorithm 4, also follows the well-known principle of optimism in the face of uncertainty. In each episode $k$, we calculate the optimistic MDP $\mathcal{M}^k$ with maximum expected gain $\rho^*_{\mathcal{M}^k}$. We execute the optimal policy of $\mathcal{M}^*$ to interact with the environment and collect more samples. Once we have collected enough samples in episode $k$, we update the model class $\mathcal{U}_k$ and compute the optimistic MDP for episode $k + 1$.

Compared with the setting of episodic MDP with general function approximation (Ayoub et al., 2020; Wang et al., 2020; Jin et al., 2021), the additional problem in the infinite-horizon setting is that the regret technically has linear dependence on the number of total episodes, or the number of steps that we update the optimistic model and the policy. This corresponds to the last term ($KD$) in Inq 32. Therefore, to design efficient algorithm with near-optimal regret in the infinite-horizon setting, the algorithm should maintain low-switching property (Bai et al., 2019; Kong et al., 2021). Taking inspiration from the recent work that studies efficient exploration with low switching cost

in episodic setting (Kong et al., 2021), we define the importance score, $\sup_{f_1, f_2 \in \mathcal{F}} \frac{\|f_1 - f_2\|^2_{\mathcal{Z}_{new}}}{\|f_1 - f_2\|^2_{\mathcal{Z}} + \alpha}$, as a measure of the importance for new samples collected in current episode, and only update the optimistic model and the policy when the importance score is greater than 1. Here $\|f_1 - f_2\|^2_{\mathcal{Z}}$ is a shorthand of $\sum_{(s,a,s',\lambda) \in \mathcal{Z}} (f_1(s, a, \lambda) - f_2(s, a, \lambda))^2$.

---

**Algorithm 4** General Optimistic Algorithm

---

1: Initialize: the MDP set $\mathcal{U}_1 = \mathcal{U}$, episode counter $k = 1$
2: Initialize: the history data set $\mathcal{Z} = \emptyset$, $\mathcal{Z}_{new} = \emptyset$
3: $\alpha = 4D^2 + 1$, $\beta = cD^2 \log(H \cdot \mathcal{N}(\mathcal{F}, 1/H))$ for a constant $c$.
4: Compute $\mathcal{M}^1 = \arg\max_{\mathcal{M} \in \mathcal{U}_1} \rho^\star_{\mathcal{M}}$
5: **for** step $h = 1, \cdots, H$ **do**
6:     Take action $a_h = \pi^*_{\mathcal{M}^k}(s_h)$ in the current state $s_h$, and transit to state $s_{h+1}$
7:     Add $(s_h, a_h, s_{h+1}, \lambda^*_{\mathcal{M}^k})$ to the set $\mathcal{Z}_{new}$
8:     **if** importance score $\sup_{f_1, f_2 \in \mathcal{F}} \frac{\|f_1 - f_2\|^2_{\mathcal{Z}_{new}}}{\|f_1 - f_2\|^2_{\mathcal{Z}} + \alpha} \geq 1$ **then**
9:         Add the history data $\mathcal{Z}_{new}$ to the set $\mathcal{Z}$
10:         Calculate $\hat{\mathcal{M}}_{k+1} = \arg\min_{\mathcal{M} \in \mathcal{U}} \sum_{(s_h, a_h, s_{h+1}, \lambda_h) \in \mathcal{Z}} (P_{\mathcal{M}} \lambda_h(s_h, a_h) - \lambda_h(s_{h+1}))^2$
11:         Update $\mathcal{U}_{k+1} = \left\{ \mathcal{M} \in \mathcal{U} : \sum_{(s_h, a_h, s_{h+1}, \lambda_h) \in \mathcal{Z}} \left( \left( P_{\mathcal{M}} - P_{\hat{\mathcal{M}}_{k+1}} \right) \lambda_h(s_h, a_h) \right)^2 \leq \beta \right\}$
12:         Compute $\mathcal{M}^{k+1} = \arg\max_{\mathcal{M} \in \mathcal{U}_{k+1}} \rho^\star_{\mathcal{M}}$
13:         Episode counter $k = k + 1$
14:     **end if**
15: **end for**

---

We state the regret upper bound of Algorithm 4 in Theorem 5, and defer the proof of the theorem to Appendix F.1.

**Theorem 7.** *Under Assumption 1, the regret of Algorithm 4 is uppder bounded by*

$$Reg(H) \leq O\left( D\sqrt{d_e H \log(H \cdot \mathcal{N}(\mathcal{F}, 1/H))} \right), \tag{35}$$

*where $d_e$ is the $\epsilon$-eluder dimension of function class $\mathcal{F}$ with $\epsilon = \frac{1}{H}$, and $\mathcal{N}(\mathcal{F}, 1/H)$ is the $\frac{1}{H}$-covering number of $\mathcal{F}$ w.r.t $L_\infty$ norm.*

For $\alpha > 0$, we say the covering number $\mathcal{N}(\mathcal{F}, \alpha)$ of $\mathcal{F}$ w.r.t the $L_\infty$ norm equals $m$ if there is $m$ functions in $\mathcal{F}$ such that any function in $\mathcal{F}$ is at most $\alpha$ away from one of these $m$ functions in norm $\| \cdot \|_\infty$. The $\| \cdot \|_\infty$ norm of function $f$ is defined as $\|f\|_\infty \stackrel{\text{def}}{=} \max_{x \in \mathcal{X}} |f(x)|$.

## D    OMITTED PROOF FOR FINITE SIMULATOR CLASS WITH SEPARATION CONDITION

### D.1    PROOF OF THEOREM 5

The formal definition of $\hat{\pi}$ is given in Algorithm 1. To upper bound the sim-to-real gap of $\hat{\pi}$, we discuss on the following two benign properties of Algorithm 1 in Lemma 4 and Lemma 7. Lemma 4 states that the true MDP $\mathcal{M}^*$ will never be eliminated from the MDP set $\mathcal{D}$. Therefore, in stage 2, the agent will execute the optimal policy in the remaining steps with high probability. Lemma 7 states that the total number of steps in stage 1 is upper bounded by $\tilde{O}(\frac{M^2}{\delta^4})$. This is where the final bound in Theorem 5 comes from.

**Lemma 4.** *With probability at least $1 - \frac{1}{H}$, the true MDP $\mathcal{M}^*$ will never be eliminated from the MDP set $\mathcal{D}$ in stage 1.*

The while-loop in stage 1 will last for $M - 1$ times. To prove Lemma 4, we need to prove that, if the true MDP $\mathcal{M}^*$ is selected in a certain loop, then $\prod_{(s,a,s') \in \mathcal{H}_{\mathcal{M}_1, \mathcal{M}_2}} \frac{P_{\mathcal{M}^*}(s'|s,a)}{P_{\mathcal{M}}(s'|s,a)} \geq 1$ holds with high probability. This is illustrated in the following lemma.

**Lemma 5.** *Suppose $\mathcal{H} = \{s_i'\}_{i=1}^{n_0}$ is a set of $n_0 = \frac{c_0 \log^2(SMH/\delta) \log(MH)}{\delta^4}$ independent samples from a given state-action pair $(s_0, a_0)$ and MDP $\mathcal{M}^*$ for a large constant $c_0$. Let $\mathcal{M}_1$ denote another MDP satisfying $\|(P_{\mathcal{M}^*} - P_{\mathcal{M}_1})(\cdot|s_0, a_0)\|_1 \geq \delta$, then the following inequality holds with probability at least $1 - \frac{1}{MH}$:*

$$\prod_{s' \in \mathcal{H}} \frac{P_{\mathcal{M}^*}(s'|s_0, a_0)}{P_{\mathcal{M}_1}(s'|s_0, a_0)} > 1, \tag{36}$$

*Proof.* The proof of Lemma 5 is inspired by the analysis in Kwon et al. (2021). To prove Inq 36, it is enough to show that

$$\ln \left( \prod_{s' \in \mathcal{H}} \frac{P_{\mathcal{M}^*}(s'|s_0, a_0)}{P_{\mathcal{M}_1}(s'|s_0, a_0)} \right) = \sum_{s' \in \mathcal{H}} \ln \left( \frac{P_{\mathcal{M}^*}(s'|s_0, a_0)}{P_{\mathcal{M}_1}(s'|s_0, a_0)} \right) > 0. \tag{37}$$

holds with probability at least $1 - \frac{1}{MH}$.

Note that $\frac{P_{\mathcal{M}^*}(s'|s_0, a_0)}{P_{\mathcal{M}_1}(s'|s_0, a_0)}$ can be unbounded since $P_{\mathcal{M}_1}(s'|s_0, a_0)$ can be zero for certain $s'$. To tackle this issue, we define $\tilde{P}_{\mathcal{M}_1}$ for a sufficiently small $\alpha \leq \frac{\delta}{4S}$:

$$\tilde{P}_{\mathcal{M}_1}(s'|s, a) = \alpha + (1 - \alpha S)P_{\mathcal{M}_1}(s'|s, a). \tag{38}$$

We have $\left\| \left( \tilde{P}_{\mathcal{M}_1} - P_{\mathcal{M}_1} \right)(\cdot|s, a) \right\| \leq 2S\alpha \leq \frac{\delta}{2}$, thus $\left\| \left( \tilde{P}_{\mathcal{M}_1} - P_{\mathcal{M}^*} \right)(\cdot|s, a) \right\| \geq \frac{\delta}{2}$. Also, we have $\ln \left( \frac{1}{\tilde{P}_{\mathcal{M}_1}(s'|s_0, a_0)} \right) \leq \ln(1/\alpha)$ for any $s' \in \mathcal{S}$. With the above definition, we can decompose Inq 37 into two terms:

$$\sum_{s' \in \mathcal{H}} \ln \left( \frac{P_{\mathcal{M}^*}(s'|s_0, a_0)}{P_{\mathcal{M}_1}(s'|s_0, a_0)} \right) = \sum_{s' \in \mathcal{H}} \ln \left( \frac{P_{\mathcal{M}^*}(s'|s_0, a_0)}{\tilde{P}_{\mathcal{M}_1}(s'|s_0, a_0)} \right) + \sum_{s' \in \mathcal{H}} \ln \left( \frac{\tilde{P}_{\mathcal{M}_1}(s'|s_0, a_0)}{P_{\mathcal{M}_1}(s'|s_0, a_0)} \right). \tag{39}$$

Taking expectation over $s' \sim P_{\mathcal{M}^*}(\cdot|s, a)$ for the first term, we have

$$\mathbb{E}\left[ \sum_{i=1}^{n_0} \ln \left( \frac{P_{\mathcal{M}^*}(s_i'|s_0, a_0)}{\tilde{P}_{\mathcal{M}_1}(s_i'|s_0, a_0)} \right) \right] = \sum_{i=1}^{n_0} \sum_{s'} P_{\mathcal{M}^*}(s'|s_0, a_0) \ln \left( \frac{P_{\mathcal{M}^*}(s'|s_0, a_0)}{\tilde{P}_{\mathcal{M}_1}(s'|s_0, a_0)} \right) \tag{40}$$

$$= n_0 D_{KL} \left( P_{\mathcal{M}^*}(s'|s_0, a_0) | \tilde{P}_{\mathcal{M}_1}(s'|s_0, a_0) \right) \tag{41}$$

$$\geq \frac{n_0 \delta^2}{2}, \tag{42}$$

where the last inequality is due to Pinsker's inequality.

**Lemma 6.** *(Lemma C.2 in Kwon et al. (2021)) Suppose $X$ is an arbitrary discrete random variable on a finite support $\mathcal{X}$. Then, $\ln(1/P(X))$ is a sub-exponential random variable (Vershynin, 2010) with Orcliz norm $\ln(1/P(X))_{\phi_1} = 1/e$.*

From the above Lemma, we know that both $\tilde{P}_{\mathcal{M}_1}(s'|s_0, a_0)$ and $P_{\mathcal{M}^*}(s'|s_0, a_0)$ are sub-exponential random variables. By Azuma-Hoeffing's inequality, we have with probability at least $1 - \delta_0/2$,

$$\sum_{s' \in \mathcal{H}} \ln \left( \frac{1}{\tilde{P}_{\mathcal{M}_1}(s'|s_0, a_0)} \right) \geq \mathbb{E}\left[ \sum_{i=1}^{n_0} \ln \left( \frac{1}{\tilde{P}_{\mathcal{M}_1}(s_i'|s_0, a_0)} \right) \right] - \log(1/\alpha)\sqrt{2n_0 \log(2/\delta_0)}. \tag{43}$$

By Proposition 5.16 in Vershynin (2010), with probability at least $1 - \delta_0/2$,

$$\sum_{s' \in \mathcal{H}} \ln \left( P_{\mathcal{M}^*}(s'|s_0, a_0) \right) \geq \mathbb{E}\left[ \sum_{i=1}^{n_0} \ln \left( P_{\mathcal{M}^*}(s_i'|s_0, a_0) \right) \right] - \sqrt{n_0 \log(2/\delta_0)/c}, \tag{44}$$

for a certain constant $c > 0$. Therefore, we can lower bound the first term in Eqn 39,

$$\sum_{s' \in \mathcal{H}} \ln \left( \frac{P_{\mathcal{M}^*}(s'|s_0, a_0)}{\tilde{P}_{\mathcal{M}_1}(s'|s_0, a_0)} \right) \geq \frac{n_0 \delta^2}{2} - \log(1/\alpha) \sqrt{2n_0 \log(2/\delta_0)} - \sqrt{n_0 \log(2/\delta_0)/c}, \quad (45)$$

with probability at least $1 - \delta_0$.

For the second term in Eqn 39, by the definition of $\tilde{P}_{\mathcal{M}_1}$,

$$\sum_{s' \in \mathcal{H}} \ln \left( \frac{\tilde{P}_{\mathcal{M}_1}(s'|s_0, a_0)}{P_{\mathcal{M}_1}(s'|s_0, a_0)} \right) \geq -2\alpha S n_0 \quad (46)$$

Combining Inq 45 and Inq 46, we have

$$\sum_{s' \in \mathcal{H}} \ln \left( \frac{P_{\mathcal{M}^*}(s'|s_0, a_0)}{P_{\mathcal{M}_1}(s'|s_0, a_0)} \right) \geq \frac{n_0 \delta^2}{2} - \log(1/\alpha) \sqrt{2n_0 \log(2/\delta_0)} - \sqrt{n_0 \log(2/\delta_0)/c} - 2\alpha S n_0 \quad (47)$$

Setting $\alpha = \frac{\delta^2}{8S}$, $\delta_0 = \frac{1}{MH}$ , and $n_0 = \frac{c_0 \log^2(SMH/\delta) \log(MH)}{\delta^4}$, we have

$$\sum_{s' \in \mathcal{H}} \ln \left( \frac{P_{\mathcal{M}^*}(s'|s_0, a_0)}{P_{\mathcal{M}_1}(s'|s_0, a_0)} \right) > 0 \quad (48)$$

holds with probability at least $1 - \frac{1}{MH}$. $\qquad \square$

**Lemma 7.** *Suppose Stage 1 in Algorithm 1 ends in $h_0$ steps. We have $\mathbb{E}[h_0] \leq O(\frac{DM^2 \log^2(SMH/\delta) \log(MH)}{\delta^4})$, where the expectation is over all randomness in the algorithm and the environment.*

*Proof.* Recall that $\pi_{\mathcal{M}}(s, s')$ is the policy with the minimum expected travelling time $\mathbb{E}[T(s' \mid \mathcal{M}, \pi, s)]$ for MDP $\mathcal{M}$. By Assumption 1, we have

$$\mathbb{E}[T(s' \mid \mathcal{M}^*, \pi_{\mathcal{M}}(s, s'), s)] \leq D. \quad (49)$$

Given state $s$ and $s'$, by Markov's inequality, we know that with probability at least $\frac{1}{2}$,

$$T(s' \mid \mathcal{M}^*, \pi_{\mathcal{M}^*}(s, s'), s) \leq 2D. \quad (50)$$

Consider the following stochastic process: In each episode $k$, the agent starts from a state $s_k$ that is arbitrarily selected, and run the policy $\pi_{\mathcal{M}^*}(s_k, s_0)$ for $2D$ steps on the MDP $\mathcal{M}^*$. The process terminates once the agent enters a certain state $s_0$. By Inq 50, the probability that the process terminates within $k$ episodes is at least $1 - \frac{1}{2^k}$. By the basic algebraic calculation, the expected stopping episode can be bounded by a constant 4.

Now we return to the proof of Lemma 7. In Subroutine 2, we run policy $\pi_{\mathcal{M}_i}$ for each MDP $\mathcal{M}_i \in \mathcal{D}$ alternately. By Lemma 4, the true MDP $\mathcal{M}^*$ is always contained in the MDP set $\mathcal{D}$. Therefore, the expected travelling time to enter state $s_0$ for $n_0$ times is bounded by $n_0 \cdot M \cdot 8D$. In stage 1, we call Subroutine 2 for $M - 1$ times, which means that the expected steps in stage 1 satisfies $\mathbb{E}[h_0] \leq 8M^2 n_0 d = O(\frac{DM^2 \log^2(SMH/\delta) \log(MH)}{\delta^4})$. $\qquad \square$

*Proof.* (Proof of Theorem 5) Recall that we use $h_0$ to denote the total steps in stage 1. Firstly, we prove that $\text{Gap}(\hat{\pi}, \mathcal{U})$ is upper bounded by $O\left(\mathbb{E}[h_0] + D\right)$.

$$V^*_{\mathcal{M}^*,1}(s_1) - V^{\hat{\pi}}_{\mathcal{M}^*,1}(s_1) \tag{51}$$

$$= \mathbb{E}_{h_0}\left[\mathbb{E}_{\mathcal{M}^*,\pi^*_{\mathcal{M}^*}}\left[\sum_{h=1}^{h_0} R(s_h, a_h) \mid h_0\right] + \mathbb{E}_{\mathcal{M}^*,\pi^*_{\mathcal{M}^*}}\left[V^*_{\mathcal{M}^*,h_0+1}(s_{h_0+1}) \mid h_0\right]\right] \tag{52}$$

$$- \mathbb{E}_{h_0}\left[\mathbb{E}_{\mathcal{M}^*,\hat{\pi}}\left[\sum_{h=1}^{h_0} R(s_h, a_h) \mid h_0\right] + \mathbb{E}_{\mathcal{M}^*,\hat{\pi}}\left[V^{\hat{\pi}}_{\mathcal{M}^*,h_0+1}(s_{h_0+1}) \mid h_0\right]\right] \tag{53}$$

$$\leq \mathbb{E}_{h_0}\left[\mathbb{E}_{\mathcal{M}^*,\pi^*_{\mathcal{M}^*}}\left[\sum_{h=1}^{h_0} R(s_h, a_h) \mid h_0\right]\right] + \mathbb{E}_{h_0}\left[\mathbb{E}_{\mathcal{M}^*,\pi^*_{\mathcal{M}^*}}\left[V^*_{\mathcal{M}^*,h_0+1}(s_{h_0+1}) \mid h_0\right] - \mathbb{E}_{\mathcal{M}^*,\hat{\pi}}\left[V^{\hat{\pi}}_{\mathcal{M}^*,h_0+1}(s_{h_0+1}) \mid h_0\right]\right] \tag{54}$$

$$\leq \mathbb{E}[h_0] + \mathbb{E}_{h_0}\left[\mathbb{E}_{\mathcal{M}^*,\pi^*_{\mathcal{M}^*}}\left[V^*_{\mathcal{M}^*,h_0+1}(s_{h_0+1}) \mid h_0\right] - \mathbb{E}_{\mathcal{M}^*,\hat{\pi}}\left[V^{\hat{\pi}}_{\mathcal{M}^*,h_0+1}(s_{h_0+1}) \mid h_0\right]\right] \tag{55}$$

The outer expectation is over all possible $h_0$, while the inner expectation is over the possible trajectories given fixed $h_0$. By Lemma 4, we know that $\hat{\pi} = \pi^*_{\text{DR}}$ after $h_0$ steps with probability at least $1 - \frac{1}{H}$. If this high-probability event happens, the second part in the above inequality equals to

$$\mathbb{E}\left[\mathbb{E}_{\mathcal{M}^*,\pi^*_{\mathcal{M}^*}}\left[V^*_{\mathcal{M}^*,h_0+1}(s_{h_0+1}) \mid h_0\right] - \mathbb{E}_{\mathcal{M}^*,\hat{\pi}}\left[V^*_{\mathcal{M}^*,h_0+1}(s_{h_0+1}) \mid h_0\right]\right].$$

We can prove that this term is upper bounded by $2D$. Given fixed $h_0$, $\mathbb{E}_{\mathcal{M}^*,\pi^*_{\mathcal{M}^*}}\left[V^*_{\mathcal{M}^*,h_0+1}(s_{h_0+1}) \mid h_0\right] - \mathbb{E}_{\mathcal{M}^*,\hat{\pi}}\left[V^*_{\mathcal{M}^*,h_0+1}(s_{h_0+1}) \mid h_0\right]$ is the difference of the value function in step $h_0 + 1$ under two different distribution of $s_{h_0+1}$. We use $d_h(s, \pi)$ to denote the state distribution in step $h$ after starting from state $s$ in step $h_0 + 1$ following policy $\pi$.

$$V^*_{\mathcal{M}^*,h_0+1}(s_{h_0+1}) = \sum_{h=h_0+1}^{H} \mathbb{E}_{s_h \sim d_h(s_{h_0+1}, \pi^*_{\mathcal{M}^*})} R(s_h, \pi^*_{\text{DR}}(s_h)) \tag{56}$$

$$= \sum_{h=h_0+1}^{H} \left(\rho^*_{\mathcal{M}^*} + \mathbb{E}_{s_h \sim d_h(s_{h_0+1}, \pi^*_{\mathcal{M}^*})}\lambda^*_{\mathcal{M}^*}(s_h) - \mathbb{E}_{s_{h+1} \sim d_{h+1}(s_{h_0+1}, \pi^*_{\mathcal{M}^*})}\lambda^*_{\mathcal{M}^*}(s_{h+1})\right) \tag{57}$$

$$= (H - h_0)\rho^*_{\mathcal{M}^*} + \lambda^*_{\mathcal{M}^*}(s_{h_0+1}) - \mathbb{E}_{s_{H+1} \sim d_{H+1}(s_{h_0+1}, \pi^*_{\mathcal{M}^*})}\lambda^*_{\mathcal{M}^*}(s_{H+1}), \tag{58}$$

where the second equality is due to the Bellman equation in infinite-horizon setting (Eqn 10). Note that by the communicating property, we have $0 \leq \lambda^*_{\mathcal{M}^*}(s) \leq D$ for any $s \in \mathcal{S}$. Therefore, we have

$$\mathbb{E}_{\mathcal{M}^*,\pi^*_{\mathcal{M}^*}}\left[V^*_{\mathcal{M}^*,h_0+1}(s_{h_0+1}) \mid h_0\right] - \mathbb{E}_{\mathcal{M}^*,\hat{\pi}}\left[V^*_{\mathcal{M}^*,h_0+1}(s_{h_0+1}) \mid h_0\right] \leq 2D. \tag{59}$$

If the high-probability event defined in Lemma 4 does not hold (This happens with probability at most $\frac{1}{H}$), we still have $\mathbb{E}_{\mathcal{M}^*,\pi^*_{\mathcal{M}^*}}\left[V^*_{\mathcal{M}^*,h_0+1}(s_{h_0+1}) \mid h_0\right] - \mathbb{E}_{\mathcal{M}^*,\hat{\pi}}\left[V^{\hat{\pi}}_{\mathcal{M}^*,h_0+1}(s_{h_0+1}) \mid h_0\right] \leq H$, thus this does not influence the final bound. Taking expectation over all possible $h_0$, and plugging the result back to Inq 51, we have

$$V^*_{\mathcal{M}^*,1}(s_1) - V^{\hat{\pi}}_{\mathcal{M}^*,1}(s_1) \leq O\left(\mathbb{E}[h_0] + D\right) = O\left(\frac{DM^2 \log^2(SMH/\delta)\log(MH)}{\delta^4}\right). \tag{60}$$

$\square$

## D.2 PROOF OF THEOREM 1

*Proof.* The theorem can be proved by combining Theorem 5 and Lemma 1.

By Theorem 5, we prove that the constructed policy $\hat{\pi}$ satisfies

$$V^*_{\mathcal{M}^*,1}(s_1) - V^{\hat{\pi}}_{\mathcal{M}^*,1}(s_1) \leq O\left(\frac{DM^2 \log^2(SMH)\log(MH)}{\delta^4}\right). \tag{61}$$

By Lemma 1, the sim-to-real gap of policy $\pi_{\text{DR}}^*$ is bounded by

$$\text{Gap}(\pi_{\text{DR}}^*, \mathcal{U}) \leq O\left(\frac{DM^3 \log^2(SMH)\log(MH)}{\delta^4}\right). \tag{62}$$

$\square$

# E   OMITTED PROOF FOR FINITE SIMULATOR CLASS WITHOUT SEPARATION CONDITION

## E.1   PROOF OF THEOREM 6

**Lemma 8.** *(Optimism) With probability at least $1 - \frac{1}{MH}$, we have $\rho_{\mathcal{M}^k}^* \geq \rho_{\mathcal{M}^*}^*$ for any $k \in [K]$.*

*Proof.* For any fixed $\mathcal{M} \in \mathcal{U}$, and fixed step $h \in [H]$, by Azuma's inequality, we have with probability at least $1 - \frac{1}{M^2H^2}$,

$$\left|\sum_{t=h_0}^h \left(\lambda_{\mathcal{M}}^*(s_{t+1}) - P_{\mathcal{M}^*}\lambda_{\mathcal{M}}^*(s_t, a_t)\right)\right| \leq D\sqrt{2(h - h_0)\log(2HM)}. \tag{63}$$

Taking union bounds over all possible $\mathcal{M}$ and $h$, we know that the above event holds for all possible $\mathcal{M}$ and $h$ with probability $1 - \frac{1}{MH}$. Under the above event, the true MDP $\mathcal{M}^*$ will never be eliminated from the MDP set $\mathcal{U}_k$. Therefore, we have $\rho_{\mathcal{M}^k}^* \geq \rho_{\mathcal{M}^*}^*$. $\square$

*Proof.* (Proof of Theorem 6) By Lemma 8, we know that $\rho_{\mathcal{M}^k}^* \geq \rho_{\mathcal{M}^*}^*$ for any $k \in [K]$. We use $\tau(h)$ to denote the episode that step $h$ belongs to. We can upper bound the regret in $H$ steps as follows:

$$H\rho_{\mathcal{M}^*}^* - \sum_{h=1}^H R(s_h, a_h) \tag{64}$$

$$\leq \sum_{h=1}^H \left(\rho_{\mathcal{M}^{\tau(h)}}^* - R(s_h, a_h)\right) \tag{65}$$

$$= \sum_{h=1}^H \left(P_{\mathcal{M}^{\tau(h)}}\lambda_{\mathcal{M}^{\tau(h)}}^*(s_h, a_h) - \lambda_{\mathcal{M}^{\tau(h)}}^*(s_h)\right) \tag{66}$$

$$= \sum_{h=1}^H \left(P_{\mathcal{M}^{\tau(h)}} - P_{\mathcal{M}^*}\right)\lambda_{\mathcal{M}^{\tau(h)}}^*(s_h, a_h) + \sum_{h=1}^H \left(P_{\mathcal{M}^*}\lambda_{\mathcal{M}^{\tau(h)}}^*(s_h, a_h) - \lambda_{\mathcal{M}^{\tau(h)}}^*(s_h)\right) \tag{67}$$

$$= \sum_{h=1}^H \left(P_{\mathcal{M}^{\tau(h)}} - P_{\mathcal{M}^*}\right)\lambda_{\mathcal{M}^{\tau(h)}}^*(s_h, a_h) + P_{\mathcal{M}^*}\lambda_{\mathcal{M}^{\tau(h)}}^*(s_{h_0}, a_{h_0}) - \lambda_{\mathcal{M}^{\tau(h)}}^*(s_1) \tag{68}$$

$$+ \sum_{h=1}^{H-1} \left(P_{\mathcal{M}^*}\lambda_{\mathcal{M}^{\tau(h)}}^*(s_h, a_h) - \lambda_{\mathcal{M}^{\tau(h)}}^*(s_{h+1})\right) \tag{69}$$

By Azuma's inequality, we know that

$$\sum_{h=1}^{H-1} \left(P_{\mathcal{M}^*}\lambda_{\mathcal{M}^{\tau(h)}}^*(s_h, a_h) - \lambda_{\mathcal{M}^{\tau(h)}}^*(s_{h+1})\right) \leq D\sqrt{H\log(2MH)}, \tag{70}$$

holds with probability at least $1 - \frac{1}{MH}$. Since $0 \leq \lambda_{\mathcal{M}}^* \leq D$, we have $P_{\mathcal{M}^*}\lambda_{\mathcal{M}^{\tau(h)}}^*(s_{h_0}, a_{h_0}) - \lambda_{\mathcal{M}^{\tau(h)}}^*(s_1) \leq D$. Therefore, we have

$$H\rho_{\mathcal{M}^*}^* - \sum_{h=1}^H R(s_h, a_h) \leq \sum_{h=1}^H \left(P_{\mathcal{M}^{\tau(h)}} - P_{\mathcal{M}^*}\right)\lambda_{\mathcal{M}^{\tau(h)}}^*(s_h, a_h) + D\sqrt{H\log(2HM)} + D \tag{71}$$

$$\leq D\sqrt{2MH\log(2MH)} + M + D\sqrt{H\log(2HM)} + D. \tag{72}$$

The first term in (71) is bounded by the stopping condition (line 5 of Algorithm 3). By Lemma 2, we have $V^*_{\mathcal{M}^*,1}(s_1) \leq H\rho^*_{\mathcal{M}^*} + D$. Therefore, we have $V^*_{\mathcal{M}^*,1}(s_1) - \sum_{h=1}^H R(s_h, a_h) \leq O\left(D\sqrt{MH\log(MH)}\right)$ with probability at least $1 - \frac{2}{MH}$. Therefore, we have

$$V^*_{\mathcal{M}^*,1}(s_1) - V^{\pi^*_{DR}}_{\mathcal{M}^*,1}(s_1) \leq O\left(D\sqrt{MH\log(MH)} + H \cdot \frac{2}{MH})\right) = O\left(D\sqrt{MH\log(MH)}\right). \tag{73}$$

$\square$

### E.2 PROOF OF THEOREM 2

Theorem 2 can be proved by combining Theorem 6 , Lemma 1 and Lemma 2. By Theorem 6, for any $\mathcal{M} \in \mathcal{U}$, the policy $\hat{\pi}$ represented by Algorithm 3 has regret bound $H\rho^*_{\mathcal{M}} - \sum_{h=1}^H R(s_h, a_h) \leq O\left(D\sqrt{MH\log(MH)}\right)$. Taking expectation over $\{s_h, a_h\}_{h\in[H]}$ and combining the inequality with Lemma 2, we have for any $\mathcal{M} \in \mathcal{U}$.

$$V^*_{\mathcal{M},1}(s_1) - V^{\hat{\pi}}_{\mathcal{M},1}(s_1) \leq O\left(D\sqrt{MH\log(MH)}\right). \tag{74}$$

Then the theorem can be proved by Lemma 1.

## F OMITTED PROOF FOR INFINITE SIMULATOR CLASS

### F.1 PROOF OF THEOREM 7

**Lemma 9.** *(Low Switching) The total number of episode $K$ is bounded by*

$$K \leq O(dim_E(\mathcal{F}, 1/H)\log(D^2H)\log(H)) \tag{75}$$

*Proof.* By Lemma 5 of Kong et al. (2021), we know that

$$\sum_{t=1}^H \min\left\{\sup_{f_1,f_2\in\mathcal{F}} \frac{(f_1(x_t) - f_2(x_t))^2}{\|f_1 - f_2\|^2_{\mathcal{Z}_t} + 1}, 1\right\} \leq C\mathrm{dim}_E(\mathcal{F}, 1/H)\log(D^2H)\log(H) \tag{76}$$

for some constant $C > 0$.

Our idea is to use this result to upper bound the number of total switching steps.

Let $\tau(k)$ be the first step of episode $k$. By the definition of the function class $\mathcal{F}$, we have $(f_1 - f_2)^2(x_t) \leq 4D^2$ for any $f_1, f_2 \in \mathcal{F}$ and $x_t$. By the switching rule, we know that, once the agent starts a new episode after step $\tau(k + 1) - 1$, we have

$$\sum_{t=\tau(k)}^{\tau(k+1)-1} (f_1 - f_2)^2(x_t) \leq \sum_{t=1}^{\tau(k)-1} (f_1 - f_2)^2(x_t) + \alpha + 4D^2, \forall f_1, f_2, x_t \tag{77}$$

Therefore, we have

$$\sum_{t=1}^{\tau(k+1)-1} (f_1 - f_2)^2(x_t) \leq 2\sum_{t=1}^{\tau(k)-1} (f_1 - f_2)^2(x_t) + \alpha + 4D^2, \forall f_1, f_2, x_t \tag{78}$$

Now we upper bound the importance score in the switching step $\tau(k+1) - 1$.

$$\min\left\{\sup_{f_1,f_2}\frac{\sum_{t=\tau(k)}^{\tau(k+1)-1}(f_1(x_t)-f_2(x_t))^2}{\|f_1-f_2\|^2_{\mathcal{Z}_{\tau(k)}}+\alpha},1\right\} \leq \min\left\{\sum_{t=\tau(k)}^{\tau(k+1)-1}\sup_{f_1,f_2}\frac{(f_1(x_t)-f_2(x_t))^2}{\|f_1-f_2\|^2_{\mathcal{Z}_{\tau(k)}}+\alpha},1\right\}$$

$$\tag{79}$$

$$\leq \min\left\{\sum_{t=\tau(k)}^{\tau(k+1)-1}\sup_{f_1,f_2}\frac{2(f_1(x_t)-f_2(x_t))^2}{\|f_1-f_2\|^2_{\mathcal{Z}_{\tau(k+1)}}-4D^2+\alpha},1\right\}$$

$$\tag{80}$$

$$\leq \min\left\{\sum_{t=\tau(k)}^{\tau(k+1)-1}\sup_{f_1,f_2}\frac{2(f_1(x_t)-f_2(x_t))^2}{\|f_1-f_2\|^2_{\mathcal{Z}_t}-4D^2+\alpha},1\right\}$$

$$\tag{81}$$

$$\leq \sum_{t=\tau(k)}^{\tau(k+1)-1}\min\left\{\sup_{f_1,f_2}\frac{2(f_1(x_t)-f_2(x_t))^2}{\|f_1-f_2\|^2_{\mathcal{Z}_t}-4D^2+\alpha},1\right\}$$

$$\tag{82}$$

Suppose the number of episodes is at most $K$. If we set $\alpha = 4D^2 + 1$, we have

$$\sum_{k=1}^{K}\min\left\{\sup_{f_1,f_2}\frac{\sum_{t=\tau(k)}^{\tau(k+1)-1}(f_1(x_t)-f_2(x_t))^2}{\|f_1-f_2\|^2_{\mathcal{Z}_{\tau(k)}}+\alpha},1\right\} \leq \sum_{t=1}^{H}\min\left\{\sup_{f_1,f_2}\frac{2(f_1(x_t)-f_2(x_t))^2}{\|f_1-f_2\|^2_{\mathcal{Z}_t}-4D^2+2\alpha},1\right\}$$

$$\tag{83}$$

$$\leq C\mathrm{dim}_E(\mathcal{F},1/H)\log(D^2H)\log(H) \tag{84}$$

By the switching rule, we have $\sup_{f_1,f_2}\frac{\sum_{t=\tau(k)}^{\tau(k+1)-1}(f_1(x_t)-f_2(x_t))^2}{\|f_1-f_2\|^2_{\mathcal{Z}_{\tau(k)}}+\alpha} \geq 1$. Therefore, the LHS of the above inequality is exactly $K$. Thus we have

$$K \leq C\mathrm{dim}_E(\mathcal{F},1/H)\log(D^2H)\log(H). \tag{85}$$

$\square$

**Lemma 10.** *(Optimism) With probability at least $1 - \frac{1}{H}$, $\mathcal{M}^* \in \mathcal{U}_k$ holds for any episode $k \in [K]$.*

*Proof.* This lemma comes directly from Theorem 6 of Ayoub et al. (2020). Define the Filtration $\mathbb{F} = (\mathbb{F}_h)_{h>0}$ so that $\mathbb{F}_{h-1}$ is generated by $(s_1, a_1, \lambda_1, \cdots, s_h, a_h, \lambda_h)$. Then we have $\mathbb{E}[\lambda_h(s_{h+1}) \mid \mathbb{F}_{h-1}] = P_{\mathcal{M}^*}\lambda_h(s_h, a_h) = f_{\mathcal{M}^*}(s_h, a_h, \lambda_h)$. Meanwhile, $\lambda_h(s_{h+1}) - f_{\mathcal{M}^*}(s_h, a_h, \lambda_h)$ is conditionally $\frac{D}{2}$-subgaussian given $\mathbb{F}_{h-1}$. By Theorem 6 of Ayoub et al. (2020), we can directly know that $f_{\mathcal{M}^*} \in \mathcal{U}_k$ for any $k \in [K]$ with probability at least $1 - \frac{1}{H}$. $\square$

*Proof.* (Proof of Theorem 7) Let $\tau(k)$ be the first step of episode $k$. Under the high-probability event defined in Lemma 10, we can decompose the regret using the same trick in previous sections.

$$\sum_{k=1}^{K} \sum_{h=\tau(k)}^{\tau(k+1)-1} H\rho_{\mathcal{M}^*}^{\star} - R(s_h, a_h) \tag{86}$$

$$\leq \sum_{k=1}^{K} \sum_{h=\tau(k)}^{\tau(k+1)-1} (\rho_{\mathcal{M}^k}^{\star} - R(s_h, a_h)) \tag{87}$$

$$= \sum_{k=1}^{K} \sum_{h=\tau(k)}^{\tau(k+1)-1} (P_{\mathcal{M}^k}\lambda_h(s_h, a_h) - \lambda_h(s_h)) \tag{88}$$

$$= \sum_{k=1}^{K} \sum_{h=\tau(k)}^{\tau(k+1)-1} (P_{\mathcal{M}^k} - P_{\mathcal{M}^*})\lambda_h(s_h, a_h) + \sum_{k=1}^{K} \sum_{h=\tau(k)}^{\tau(k+1)-1} (P_{\mathcal{M}^*}\lambda_h(s_h, a_h) - \lambda_h(s_h)) \tag{89}$$

$$\leq \sum_{k=1}^{K} \sum_{h=\tau(k)}^{\tau(k+1)-1} (P_{\mathcal{M}^k} - P_{\mathcal{M}^*})\lambda_h(s_h, a_h) + \sum_{k=1}^{K} \sum_{h=\tau(k)}^{\tau(k+1)-2} (P_{\mathcal{M}^*}\lambda_h(s_h, a_h) - \lambda_h(s_{h+1})) + DK, \tag{90}$$

where the first inequality is due to optimism condition in Lemma 10. The first equality is due to the Bellman equation 10 and $\lambda_h = \lambda_{\mathcal{M}^k}^*$ for $\tau(k) \leq h \leq \tau(k+1) - 1$. The last inequality is due to $0 \leq \lambda_h(s) \leq D$ for any $s \in \mathcal{S}$.

Now we bound the first two terms in Eqn 90. The second term can be regarded as a martingale difference sequence. By Azuma's inequality, with probability at least $1 - \frac{1}{H}$,

$$\sum_{k=1}^{K} \sum_{h=\tau(k)}^{\tau(k+1)-1-1} (P_{\mathcal{M}^*}\lambda_h(s_h, a_h) - \lambda_h(s_{h+1})) \leq D\sqrt{2H\log(H)}. \tag{91}$$

Now we focus on the upper bound of the first term in Eqn 90. Under the high-probability event defined in Lemma 10, the true model $P$ is always in the model class $\mathcal{U}_k$. For episode $k$, from the construction of $\mathcal{U}_k$ we know that any $f_1, f_2 \in \mathcal{U}_k$ satisfies $\|f_1 - f_2\|_{\mathcal{Z}_{\tau(k)}}^2 \leq 2\beta$. Since $\mathcal{M}^k, \mathcal{M}^* \in \mathcal{U}_k$, we have

$$\sum_{t=1}^{T(k-1)} ((P_{\mathcal{M}^k} - P_{\mathcal{M}^*})\lambda_t(s_t, a_t))^2 \leq 2\beta \tag{92}$$

Moreover, by the if condition in Line 8 of Alg. 4, we have for any $\tau(k) \leq h \leq \tau(k+1) - 1$,

$$\sum_{t=\tau(k)}^{h} ((P_{\mathcal{M}^k} - P_{\mathcal{M}^*})\lambda_t(s_t, a_t))^2 \leq 2\beta + \alpha + D^2. \tag{93}$$

Summing up the above two equations, we have

$$\sum_{t=1}^{h} ((P_{\mathcal{M}^k} - P_{\mathcal{M}^*})\lambda_t(s_t, a_t))^2 \leq 4\beta + \alpha + D^2. \tag{94}$$

We invoke Lemma 26 of Jin et al. (2021) by setting $\mathcal{G} = \mathcal{F} - \mathcal{F}$, $\Pi = \{\delta_x(\cdot)|x \in \mathcal{X}\}$ where $\delta_x(\cdot)$ is the dirac measure centered at $x$, $g_t = f_{\mathcal{M}^k} - f_{\mathcal{M}^*}$, $\omega = 1/H$, $\beta = 4\beta + \alpha + D^2$ and $\mu_t = \mathbf{1}\{\cdot = (s_t, a_t, \lambda_t)\}$, then we have

$$\sum_{\tau=1}^{K} \sum_{t=S(\tau)}^{T(\tau)} |(P_{\mathcal{M}^\tau} - P_{\mathcal{M}^*}) \lambda_t(s_t, a_t)| \leq O\left(\sqrt{\dim_E(\mathcal{F}, 1/H)\beta H} + \min\left(H, \dim_E(\mathcal{F}, 1/H)\right) D + H \cdot \frac{1}{H}\right) \tag{95}$$

$$= O\left(\sqrt{\dim_E(\mathcal{F}, 1/H)\beta H}\right) \tag{96}$$

Plugging the results back to Eqn 90, we have

$$\sum_{k=1}^{K} \sum_{h=\tau(k)}^{\tau(k+1)-1} H\rho^\star - R(s_h, a_h) \leq O\left(D\sqrt{H\dim_E(\mathcal{F}, 1/H)\log\left(H \cdot \mathcal{N}(\mathcal{F}, 1/H)\right)}\right) \tag{97}$$

By Lemma 2, we have $V_{\mathcal{M}^*,1}^*(s_1) \leq H\rho_{\mathcal{M}^*}^* + D$. Therefore, we have

$$V_{\mathcal{M}^*,1}^*(s_1) - \sum_{h=1}^{H} R(s_h, a_h) \leq O\left(D\sqrt{H\dim_E(\mathcal{F}, 1/H)\log\left(H \cdot \mathcal{N}(\mathcal{F}, 1/H)\right)}\right), \tag{98}$$

with probability at least $1 - \frac{2}{MH}$. If the high-probability event doesn't holds (This happens with probability at most $\frac{2}{H}$), then the gap $V_{\mathcal{M}^*,1}^*(s_1) - V_{\mathcal{M}^*,1}^{\pi_{\text{DR}}^*}(s_1)$ still can be bounded by $H$. Taking expectation over the trajectory $\{s_h\}_h$, we have

$$V_{\mathcal{M}^*,1}^*(s_1) - V_{\mathcal{M}^*,1}^{\pi_{\text{DR}}^*}(s_1) \leq O\left(D\sqrt{H\dim_E(\mathcal{F}, 1/H)\log\left(H \cdot \mathcal{N}(\mathcal{F}, 1/H)\right)} + H \cdot \frac{2}{H}\right) \tag{99}$$

$$= O\left(D\sqrt{H\dim_E(\mathcal{F}, 1/H)\log\left(H \cdot \mathcal{N}(\mathcal{F}, 1/H)\right)}\right). \tag{100}$$

$\square$

### F.2 Proof of Theorem 4

*Proof.* Theorem 4 can be proved by combining Theorem 7, Lemma 1 and Lemma 2. By Theorem 7, for any $\mathcal{M} \in \mathcal{U}$, the policy $\hat{\pi}$ represented by Algorithm 4 can obtain regret bound $H\rho_{\mathcal{M}}^* - \sum_{h=1}^{H} R(s_h, a_h) \leq O\left(D\sqrt{d_e H \log(H \cdot \mathcal{N}(\mathcal{F}, 1/H))}\right)$. Taking expectation over $\{s_h, a_h\}_{h \in [H]}$ and combining the inequality with Lemma 2, we have for any $\mathcal{M} \in \mathcal{U}$.

$$V_{\mathcal{M},1}^*(s_1) - V_{\mathcal{M},1}^{\hat{\pi}}(s_1) \leq O\left(D\sqrt{d_e H \log(H \cdot \mathcal{N}(\mathcal{F}, 1/H))}\right). \tag{101}$$

Then the theorem can be proved by Lemma 1. $\square$

## G Lower Bounds

### G.1 Proof of Proposition 1

*Proof.* Consider the following construction of $\mathcal{U}$. There are $3M + 1$ states. There are $M$ actions in the initial state $s_0$, which is denoted as $\{a_i\}_{i=1}^M$. After taking action $a_i$ in state $s_0$, the agent will transit to state $s_{i,1}$ with probability 1. In state $s_{i,1}$ for $i \in [M]$, the agent can only take action $a_0$, and then transits to state $s_{i,2}$ with probability $p_i$, and transits to state $s_{i,3}$ with probability $1 - p_i$. State $\{s_{i,2}\}_{i=1}^M$ and $\{s_{i,3}\}_{i=1}^M$ are all absorbing states. That is, the agent can only take one action $a_0$ in these states, and transits back to the current state with probability 1. The agent can only obtain reward 1 in state $s_{i,2}$ for $i \in [M]$, and all the other states have zero rewards. Therefore, if the agent knows the transition dynamics of the MDP, it should take action $a_i$ with $i = \arg\max_i p_i$ in state $s_0$.

Now we define the transition dynamics of each MDP $\mathcal{M}_i$. For each MDP $\mathcal{M}_i \in \mathcal{U}$, we have $p_i = 1$ and $p_j = 0$ for all $j \in [M], j \neq i$. Therefore, the agent cannot identify $\mathcal{M}^*$ in state $s_0$. The best policy in state $s_0$ for latent MDP $\mathcal{U}$ is to randomly take an action $a_i$. In this case, the sim-to-real gap can be at least $\Omega(H)$ since the agent takes the wrong action in state $s_0$ with probability $1 - \frac{1}{M}$.

$\square$

## G.2 PROOF OF THEOREM 3

*Proof.* We first show that $\Omega(\sqrt{MH})$ holds with the hard instance for multi-armed bandit (Lattimore & Szepesvári, 2020). Consider a class of K-armed bandits instances with $K = M$. For the bandit instance $\mathcal{M}_i$, the expected reward of arm $i$ is $\frac{1}{2} + \epsilon$, while the expected rewards of other arms are $\frac{1}{2}$. Note that this is exactly the hard instance for $K$-armed bandits. Following the proof idea of the lower bound for multi-armed bandits, we know that the regret (sim-to-real gap) is at least $\Omega(\sqrt{MH})$.

We restate the hard instance construction from Jaksch et al. (2010). This hard instance construction has also been applied to prove the lower bound in episodic setting (Jin et al., 2018). We firstly introduce the two-state MDP construction. In their construction, the reward does not depend on actions but states. State 1 always has reward 1 and state 0 always has reward 0. From state 1, any action takes the agent to state 0 with probability $\delta$, and to state 1 with probability $1 - \delta$. In state 0, there is one action $a^*$ takes the agent to state 1 with probability $\delta + \epsilon$, and the other action $a$ takes the agent to state 1 with probability $\delta$. A standard Markov chain analysis shows that the stationary distribution of the optimal policy (that is, the one that takes action $a^*$ in state 0) has a probability of being in state 1 of

$$\frac{\frac{1}{\delta}}{\frac{1}{\delta} + \frac{1}{\delta+\varepsilon}} = \frac{\delta + \varepsilon}{2\delta + \varepsilon} \geq \frac{1}{2} + \frac{\varepsilon}{6\delta} \text{ for } \varepsilon \leq \delta. \tag{102}$$

In contrast, acting sub-optimally (that is taking action $a$ in state 0) leads to a uniform distribution over the two states. The regret per time step of pulling a sub-optimal action is of order $\epsilon/\delta$.

Consider $O(S)$ copies of this two-state MDP where only one of the copies has such a good action $a^*$. These copies are connected into a single MDP with an $A$-ary tree structure. In this construction, the agent needs to identify the optimal state-action pair over totally $SA$ different choices. Setting $\delta = \frac{1}{D}$ and $\epsilon = c\sqrt{\frac{SA}{TD}}$, Jaksch et al. (2010) prove that the regret in the infinite-horizon setting is $\Omega(\sqrt{DSAT})$.

Our analysis follows the same idea of Jaksch et al. (2010). For the MDP instance $\mathcal{M}_i$, the optimal state-action pair is $(s_i, a_i)$ $((s_i, a_i) \neq (s_j, a_j), \forall i \neq j)$. With the knowledge of the transition dynamics of each $\mathcal{M}_i$, the agent needs to identify the optimal state-action pair over totally $M$ different pairs in our setting. Therefore, we can similarly prove that the lower bound is $\Omega(\sqrt{DMH})$ following their analysis. □

## G.3 LOWER BOUND IN THE LARGE SIMULATOR CLASS

**Proposition 2.** *Suppose All MDPs in the MDP set $\mathcal{U}$ are linear mixture models (Zhou et al., 2020) sharing a common low dimensional representation with dimension $d = O(\log(M))$, there exists a hard instance such that the sim-to-real gap of the policy $\pi_{DR}^*$ returned by the domain randomization oracle can be still $\Omega(H)$ when $M \geq H$.*

*Proof.* We can consider the following linear bandit instance as a special case. Suppose there are two actions with feature $\phi(a_1) = (1, 0)$ and $\phi(a_2) = (1, 1)$. In the MDP set $\mathcal{M}$, there are $M - 1$ MDPs with parameter $\theta_i = (\frac{1}{2}, -p_i)$ with $\frac{1}{4} < p_i < \frac{1}{2}, i \in [M-1]$, and one MDP with parameter $\theta_M = (\frac{1}{2}, \frac{1}{2})$. Suppose $M = 4H + 5$, the optimal policy of such an LMDP with uniform initialization will never pull the action $a_2$, which can suffer $\Omega(H)$ sim-to-real gap in the MDP with parameter $\theta_M$. □

