# OpenReview forum: "Understanding Domain Randomization for Sim-to-real Transfer"
_ICLR.cc/2022/Conference — ICLR 2022 Spotlight_

### Official Review · Reviewer_M8oh · 2021-11-01

**Correctness:** 3
**Technical Novelty And Significance:** 3
**Empirical Novelty And Significance:** Not applicable
**Recommendation:** 10
**Confidence:** 3

**Main Review:**

Let me start by admitting I am a sim-to-real practitioner, not a theorist. Therefore, it is likely that I am not fully aware of other related works that develop a similar theoretical framework. I have found the paper itself very inspiring, and both the theoretical formulation of the sim-to-real gap and the high-level idea behind the proof overview make very good sense to me. Still, I have a few comments, and it would be good if the authors could clarify them:

1. There is another possible definition of sim-to-real gaps: the performance difference between a policy $\pi$ (in particular, $\pi^*_{DR}$) in the simulation environment and in the real world:

Gap($\pi,\mathcal{U}$) = $V_{\mathcal{M}^*,1}^{\pi}(s_1) -  \mathbb{E} \_{\mathcal{M}\sim\nu} V_{\mathcal{M},1}^{\pi}(s_1)$.

While I agree Eqn. (1) is perhaps a more interesting definition of sim-to-real gaps, Eqn. (1) is fairly difficult to measure in practice. Could the authors comment on these two definitions?

2. It looks like the stepwise reward is assumed to be within 0 and 1. I guess the whole framework is also applicable to unbounded stepwise rewards, e.g., by remapping them to [0, 1] via a sigmoid function. Does this affect your main results?

3. Comparing Sec. 5.1 and Sec. 5.2 seems to imply that narrowing the sim-to-real gap is more difficult for problems without separation condition, but I have a feeling that this is not always the case. Consider an extreme scenario where all $\mathcal{M}$ in $\mathcal{U}$ are very similar to each other (and therefore they are all similar to $\mathcal{M}^*$. Such a $\mathcal{U}$ is not a separated MDP set, but I would expect the sim-to-real gap to be much smaller than the gap given for the separated MDP set (Theorem 1). In particular, if $\mathcal{U}$ simply copies $\mathcal{M}^*$ multiple times, the sim-to-real gap would be 0. It looks like assuming only a few pairs of non-separated MDPs in $\mathcal{U}$ may make the problem harder, but if all MDPs are non-separated, the problem seems to be much easier.

4. The constructive argument for proving Theorem 1 seems interesting and smart, but I have a few questions about its technical details:

* Alg. 1, line 7 (the condition in the if-statement): why not simply say “if $\Pi P\_{\mathcal{M}_1}(s’|s_0,a_0)\geq \Pi P\_{\mathcal{M}_2}(s’|s_0,a_0)$”? This would also avoid the need for treating $P\_{\mathcal{M}_2}=0$ in Sec. D.1. Also, this if-condition does not seem to be symmetric: if there exists $P\_{\mathcal{M}_1}=0$ and $P\_{\mathcal{M}_2}=0$, it always chooses to eliminate $\mathcal{M}_2$. Does it affect your proof? (Looks like it does not.)

* Eqn. (41): are $s,a$ referring to $s_0,a_0$ specifically, or are they arbitrary states and actions? It looks like they should be $s_0,a_0$, according to the sentence immediately after Eqn. (41).

* What is $\tilde{\mathcal{M}}$ in Eqn. (42) and Eqn. (49)? I was expecting $\mathcal{M}_1$ there.

* Eqn. (49): I do not see why $H$ appears on the right-hand side. It would make more sense to me if the right-hand side is $-2\alpha S n_0$.

* I cannot see how Eqn. (51) is derived from Eqn. (50) after trying to plug into Eqn. (50) the provided definitions of $\alpha$, $\delta_0$, and $n_0$. In particular, are $c$ in Eqn. (50) and $c_0$ in the definition of $n_0$ two different numbers? Also, it seems that $\delta$ in the numerator of $n_0$ between Eqn. (50) and Eqn. (51) should be replaced with $\delta_0$ so that it matches the definition of $n_0$ in Lemma 5.

A few minor comments:

1. Eqn. (1): why does Gap need to take $\mathcal{U}$ as input if all it needs is $\mathcal{M}^*$?

2. What is $d$ in Theorem 3?

3. The notation $PV$ is defined in Sec. 3.1 but does not seem to be used elsewhere in the main paper.

4. How do we check if Assumption 3 is satisfied in practice?


**Summary Of The Paper:**

This paper presents a theoretical framework for analyzing the sim-to-real gap in the context of domain randomization. The paper defines formally the sim-to-real gap and the domain randomization method. Next, it analyzes the upper bound of sim-to-real gap in three different scenarios: finite simulation classes with and without separation conditions and infinite simulation classes. The paper provides constructive arguments to prove the upper bounds.

In my opinion, this paper has made multiple contributions. I am most impressed by its theoretical framework that formally defines the sim-to-real problem and its proof of the upper bound of the sim-to-real gap based on the constructive arguments described in the paper.


**Summary Of The Review:**

I appreciate the whole theoretical framework this paper establishes for studying domain randomization techniques for sim-to-real problems. I think defining the problem formally and properly is a good contribution.

The high-level proof overview makes good sense to me. I also manually checked the proof for finite simulator class (but not very carefully) in the appendix. The whole proof seems legit despite some minor issues, which I tend to think is relatively easy to fix.

The construction of base policies (Algorithms in the appendix) may also be interesting both for sim-to-real theorists and practitioners.

Overall, I recommend strong acceptance assuming that all the equations in the main paper and in the appendix are correct or are relatively easy to fix.

---

> ### Author Response · Authors · 2021-11-22
> **Response to Reviewer M8oh**
>
> Thank you for your comments and positive feedback.
>
> **Regarding the new version of sim-to-real gap**: Thanks for the interesting idea. The sim-to-real gap defined in our paper is used as a criterion to measure the performance of a particular policy in the real environment, so the value function refers to the same (true) MDP. An upper bound of this regret for a certain policy (in particular, $\pi^*_{DR}$) directly indicates the performance of this policy in the real environment. In contrast, the different definition suggested by the reviewer characterizes the difference between the simulator class and the real environment.
> It cannot be used to measure the performance of a policy in the real world (the focus of this work), since a policy with a small gap can be arbitrarily bad in the real environment (i.e., $V^{\pi}_{\mathcal{M}^*, 1}(s_1)$ can be very small). That being said, we believe it is a potentially useful notion to study in other contexts of sim-to-real transfer.
>
> **Regrading the unbounded rewards scenario**: In most RL works, the reward function is assumed to be bounded. Any bounded reward function can always be shifted and scaled to $[0,1]$, without affecting the optimal policy [2]. Therefore, we can assume the reward is bounded in $[0,1]$ without loss of generality, as is often seen in published literature.
>
> **Regarding the hardness of problems with or without the separation condition**: The problem is easier in the extreme cases, that is, when the MDPs are very different, or when they are very similar. In the first case, it is very easy to identify them using our algorithms 1 and 2. In the second case, we do not need to identify them since we can use any of them to do planning, and the difference between them is very small. More difficult (and interesting) is the "intermediate" case, where we must do identification, and identification cannot be done trivially. We have to balance exploration (i.e. the identification of the real-world instance between different models) and exploitation (i.e. execute the policy that has high rewards without the full knowledge of the real-world instance).
>
> **Regarding the confusing statement and typos**: Thanks for the comments. We have revised the paper accordingly.
>
> **For the technical details to prove theorem 1**:
>
> - The line 7 of algorithm 1 is just for cleaner analysis, it does not affect our analysis if we change this line as you suggest. When $\mathcal{P}_{\mathcal{M}_i}=0 (i = 1, 2)$, we can eliminate any of them.
> - Eqn (41) means we modify the transition for **any $(s, a)$ pair**.
> - We are sorry about the typo in Eqn (42) and (49). It should be $\mathcal{M}_1$.
> - We are sorry about the typo in Eqn (49) and (50). It should be $-2\alpha S n_0$ in the RHS. The value of $n_0$ should be $c_0 \log^2(SMH/\delta) \log(MH) / \delta^4$ where $c_0$ is constant much larger than $c$. You can see that the derivation of Eqn (51) now is correct after we fix Eqn (50).
>
> **For the minor comments**:
>
> - The definition of $\text{Gap}$: Thanks for pointing out our mistake and we will fix the notation of the sim-to-real gap to $\text{Gap}(\pi)$, eliminating the dependence on $\mathcal{U}$.
> - The appearance of $d$ in theorem 3: Sorry for a typo. It should be $D$.
> - The $PV$ symbol: The symbol $PV$ is not used in the main context but used extensively in the appendix. We will move the definition of $PV$ to the appendix.
> - How to check if assumption 3 holds in practice: It is relative hard to check if a theoretical assumption holds in practice, as some smoothness conditions for deep neural nets are also difficult to verify. We provide an example here: consider we are exploring the effect of physical parameters (e.g., friction coeffcients, hardness of the materials, etc.) on the dexterous hand manipulation task [1], then we can set the distance between MDPs as the sum of the difference of these parameters. As we know the transition dynamic are smoothly influenced by these physical parameters, the assumption 3 naturally holds with a small Lipschitz constant.
>
> [1] OpenAI et al. Learning dexterous in-hand manipulation, *the International Journal of Robotics Research*, 2019.
>
> [2] Ng et al. Policy invariance under reward transformations: Theory and application to reward shaping, *ICML*, 1999.

---

> > ### Comment · Reviewer_M8oh · 2021-11-28
> > **Thank you for the clarification**
> >
> > **Regarding the new version of sim-to-real gap**: OK. I agree.
> >
> > **Regarding the unbounded rewards scenario**: Thank you for the reference paper.
> >
> > **Regarding the hardness of problems with or without the separation condition**: This is also what I thought when reviewing this paper. Thank you for the confirmation.
> >
> > **Regarding the confusing statement and typos**: Thank you for fixing the typos. A minor suggestion: please mark your updates to the manuscript in a different color next time. It was not easy to find these updates in the current manuscript.
> >
> > I am still willing to recommend strong acceptance for this work. However, after reading other reviewers' responses, I am afraid that none of us have managed to check all 105 equations in this work thoroughly, as can be seen from our confidence scores (Other reviewers: please feel free to correct me if I am wrong). While I have tried my best to check as many equations as I can, I kindly request the authors to check all equations one more time and make sure no typos exist.

---

### Official Review · Reviewer_y2ZB · 2021-11-03

**Correctness:** 4
**Technical Novelty And Significance:** 4
**Empirical Novelty And Significance:** Not applicable
**Recommendation:** 8
**Confidence:** 2

**Main Review:**

I found this paper very well written and easy to follow. Each assumption that the authors make - for instance that the MDPs are communicating or that the value functions are smooth - is justified by showing that there exist MDPs where the best case transfer performance is $O(H)$ in which case not much can be said. The bounds themselves are also intuitive and explain (to some extent) why zero-shot transfer is possible when training with domain randomization.

One suggestion that I had: the authors mention in the abstract and introduction that their results explain why policies trained with domain randomization should have memory. I think it might be worth mentioning in the body of the paper that the way in which Algorithms 1/3/4 use memory is somewhat different from parameterizing a network by an RNN. In these algorithms, memory is simply used to isolate the test time MDP, after which point the optimal policy (which is memoryless) for that MDP is used. In contrast, people in general would train a single RNN over the distribution of MDPs and then test the RNN on the test-time MDP. It's possible that the RNN internally learns to switch from using memory to becoming memoryless once enough observations have passed to internally determine the MDP, but it might be worth including some discussion of how Algorithms 1/3/4 compare to standard training schemes for applying domain randomization.

Currently, I feel that the bounds derived in the paper largely justify standard practices (e.g. parameterizing policies with memory when training with domain randomization, or why domain randomization should be used in the first place if one wants to transfer a policy trained in simulation to the real world), but don't suggest new practices. In the conclusion, the authors say they hope their analysis will lead to the design of more efficient algorithms for sim-to-real transfer, but I think the paper could benefit from more discussion of which of their results the believe could be the most applicable to improving how domain randomization is used in practical settings.

**Summary Of The Paper:**

The goal of this paper is formalize and analyze the technique of domain randomization in robotics. The authors frame domain randomization as training over a set of "plausible MDPs", only one of which is the MDP that will be used at test time. The authors then analyze the best possible performance of models for this problem under 3 different settings: when the set of plausible MDPs is finite with and without a separation condition, and when the set of plausible MDPs is infinite. The authors find algorithms that achieve a performance gap that beats the worst case performance gap of $O(H)$ in all three settings ($O(log^3H)$ and $O(\sqrt{H})$).

**Summary Of The Review:**

This paper formalizes the common practice of "domain randomization" and develops algorithms and bounds for the performance of domain randomization depending on if the set of randomized MDPs is finite (with or without separability) or infinite. The central result is that the performance gap is $\tilde{O}(\sqrt{H})$ in all cases. The formalization of domain randomization is accurate and the analysis of the problem setting is thorough.

---

> ### Author Response · Authors · 2021-11-22
> **Response to Reviewer y2ZB**
>
> Thanks for your comments and assessment.
>
> **Regarding the incorrect use of memory**: For algorithm 1 (finite MDPs with separation condition), we use the memory as mentioned in the review: we do isolate the test time MDP first and execute the optimal memory-less policy. However, the last two of our algorithms are using the memory as an RNN does as you have suggested in your review. We build confidence sets and eliminate unlikely MDPs with memory, and use the best policy of an MDP that is likely to be the real-world instance. In all the three settings, we deploy our algorithms 1/3/4 as a memory-based policy on the test time MDP.
>
> **Regarding the lack of suggestions of new algorithms in practice**: The reviewer is right that our work aims to provide a theoretical justification of sim-to-real transfer and domain randomization. we believe that our theory provides new insights (e.g., the importance of randomization and memory) on sim-to-real transfer, and may inspire the improvements of domain randomization or new algorithms in this research direction. We appreciate the reviewer's great suggestion, and will try to include a discussion of how our results may inspire algorithmic improvements.

---

> > ### Comment · Reviewer_y2ZB · 2021-11-28
> > **Thank you for the response**
> >
> > I appreciate the author's feedback and I'll maintain my score.
> > I think "we believe that our theory provides new insights" is somewhat of an exaggeration given that domain randomization and memory are already widely used in practice. However, I believe that rigorously demonstrating the importance of these strategies analytically adds a backbone to the intuitions that practitioners usually rely on to justify these kinds of choices, and that demonstration is a significant contribution that should be highlighted at ICLR.

---

### Official Review · Reviewer_SVo4 · 2021-11-03

**Correctness:** 4
**Technical Novelty And Significance:** 2
**Empirical Novelty And Significance:** Not applicable
**Recommendation:** 5
**Confidence:** 2

**Main Review:**

Overall I believe that the main strength of this work is providing a theoretical analysis of sim-to-real deployment via domain randomization. Sim-to-real is of particular importance for enabling safe training in simulation. Theoretically analyzing and understanding this setting has the potential to provide safety guarantees and inform the design of new algorithms. While the analysis provided by this work is interesting, in the current stage I have three major concerns. First, its motivation and contribution seem to be detached from sim-to-real practice. Second, the assumptions (if I understood them correctly) seem to be very restrictive, and finally there is no evaluation whatsoever (e.g. to quantify a gap between the derived bounds and any realistic sim-to-real setting)

# Motivation and Contribution
It is a very interesting approach to analyse sim-to-real transfer by interpreting simulators as a class of MDPs with different transition models. However, in the way this work is currently phrased, I am not sure if it provides any practically useful results. A theory work does not necessarily have to be at a stage that it is useful but I would have expected a clear technical discussion of the papers limitations in order to make use of such results in any real-world system. Currently, the link to practical implementations is provided to some extent in Sec 3.2. However, this section feels very detached from the rest of the work. It does introduce how domain randomization works in practice but does not discuss how the results of this work relate to it.

# Assumptions
The initial discussion speaks of domain randomization in general. However, the assumptions seem to be very restrictive:

* While I believe that assumption 1 is common, it would be interesting to see for which simulator real-world simulator it holds. In the case where the state-space is continuous with continuous support, the expectation $E(T(s'|\mathcal{M}, \pi, s))$ can easily become infinite for many interesting system classes?
* There is a similar issue with assumption 2. As I understand it, it makes it very hard for the simulator instances in $\mathcal{U}$ to be configurable in a continuous way because this assumption basically enforces to have a minimum distance between any two simulator instances. I wonder where this happens in practice.

Again, in theoretical work, it may be okay to have such restrictive assumptions. However, the authors should spell out which simulator & environment combinations currently satisfy such assumptions. And if there are none (which may be okay for an otherwise interesting theoretical work), there should be at least a discussion on what are the main limiting factors of the assumption compared to practical settings.

# Evaluation
I believe that the absence of any experiments is also problematic. Theoretical papers do not necessarily always need to have experiments (some theoretical contributions cannot be evaluated or may not require experimental evaluation). However, in the present case, the work aspires to address a practically relevant setting and seeing the performance evaluated seems quite relevant given that Domain Randomization already has an extensive empirical body of work. Even some of the pure theoretical results can be nicely illustrated with numerical experiments such as demonstrating cases in which derived bounds are particularly good or particularly bad.

**Summary Of The Paper:**

This paper presents a theoretical framework for reasoning and analyzing domain-randomization techniques. Its approach models a simulator instance as a Markov Decision Process for which the parameters of the MDP correspond to tunable simulator parameters. In this designed setting, the work proves sharp bounds on the gap between an optimal policy's value in the domain-randomized and the real-world setting. The work also analyzes the conditions under which sim-to-real transfer can be successful in the considered theoretical setting.

**Summary Of The Review:**

In summary, I believe that this work addresses an interesting problem. Its main shortcomings are the lack of discussion linking the particular contribution and assumptions to particular use-cases as well as the absence of any experimental demonstrations.  Addressing these shortcomings will result in an interesting contribution.

---

> ### Author Response · Authors · 2021-11-22
> **Response to Reviewer SVo4**
>
> Thank you for your suggestions and comments.
>
> **Regarding the motivation, contribution and the absence of experiments**: Our work aims to provide theoretical understanding on the mechanism how sim-to-real transfer and domain randomization work. Both of them have already seen many empirical successes (e.g., [1]), but lack principled theoretical justification.
> We develop a theoretical framework, and establish a series of theoretical results under this framework that relate to previous empirical observations. For example, our theory highlights that the success of domain randomization relies on two critical techniques, uniform randomization and the use of memory.
> While proposing new algorithms is outside the scope of the paper, our theoretical findings are useful to understand the empirical success of  these practically popular techniques. We also hope the results will inspire development of more efficient algorithms in the future.
> Finally, given the scope and theoretical nature of this work, we believe experiments are not necessary.
>
> **Regarding the assumptions**: Our assumptions are standard and common in the RL theory literature, with many examples satisfying assumptions 1 and 2. For Assumption 1, we can consider the example of robot control. In a majority of experiment setups, robotic arms can be moved to any position or changed to any joint angle within a small amount of  time, which satisfies the diameter assumption. Continuous state space is not an issue, since we can simply perform  discretization in these applications to make diameter assumption hold.
> We also provided a lower bound showing that domain randomization can fail without the Assumption 1. For Assumption 2, it holds for the dexterous hand manipulation [1] if we randomize on a discrete set of friction coefficients. Moreover, our assumption is much weaker than the separation condition proposed by [2], as we only need separation on one state-action pair. Finally, we want to point out that we also proved in the paper that domain randomization works on the last two settings without assumption 2. We agree that assumption 2 seems to be hard to satisfy for continuous simulator class as you have mentioned in the review, but we need assumption 2 only in the finite simulator class setting while continuous simulator class belongs to the infinite setting.
>
> [1] OpenAI et al. Learning dexterous in-hand manipulation, *the International Journal of Robotics Research*, 2019.
>
> [2] Kwon et al. RL for latent MDPs: Regret guarantees and a lower bound, *NeurIPS*, 2021.
>
> [3] Jaksch et al. Near-optimal regret bounds for reinforcement learning, *Journal of Machine Learning Research*, 2010.

---

> > ### Comment · Reviewer_SVo4 · 2021-11-25
> > **Response to the Authors**
> >
> > First of all, thank you for responding to my questions and being upfront not only about the strengths but also about the weaknesses of your method.
> >
> > Please allow me to respectfully disagree with your opinion on experiments. I really appreciate the theoretical work and can see how in a purely theoretical conference experiments might not be required (although even there, this seems to be less and less the case).
> >
> > My worry is that it is currently hard to quantify the extent to which the theoretical results in this work relate to any practical setting. I can think of two ways of approaching this: First, by providing additional theoretical results relating this work with real-world sim-to-real settings. Or, second, by conceiving and implementing some baseline experiments. The former is probably beyond the scope of this work even if it was rejected. However, implementing some examples showcasing the theory should, in my opinion, be doable.
> >
> > That being said, I want to reiterate the importance of theoretical work and believe that it can be a very strong contribution if linked to a traditional domain randomization setting.

---

> > > ### Author Response · Authors · 2021-11-26
> > > **Further Response**
> > >
> > > We thank the reviewer for the fast responses. We would like to clarify some potential misunderstandings about the main results of this paper and our previous response.
> > >
> > > We first would like to remark that this paper is directly related to sim-to-real in practice. The algorithm we study is precisely the domain randomization algorithm used in practical setup [1, 2]. The only minor difference is that while PPO or certain policy optimization algorithms are used for domain randomization in practice, we consider the idealized setting where those optimizers are able to return the maximizer (Definition 1). We remark that this is a reasonable idealization/approximation for what happens in practice since (1) despite that optimization of neural network (nonconvex objective) is in general NP-hard, there are many empirical evidences that SGD is able to find very good solution/minimizer in many real-world scenarios [3, 4]; (2) domain randomization would clearly fail when PPO/policy optimizer returns a bad solution. We also note that all other assumptions in this paper are not for the algorithms, but for the settings where the algorithm enjoy good theoretical guarantees. As argued in our previous response, we believe these technical assumptions are mild, standard, necessary to some extent, and applicable to real world settings in [1, 2]. Therefore, we kindly disagree with the comment that this work needs to "provide additional theoretical results relating this work with real-world sim-to-real setting".
> > >
> > > Second, since the algorithm studied in this work is just the domain randomization used in practice, empirical works [1, 2] already provide extensive experiments under various settings. We believe that running experiments of domain randomization on toy settings would not add much value to justify the power of the algorithm given these existing extensive empirical studies. Therefore, we opt to focus on providing theoretical understandings and justifications.
> > >
> > > [1] OpenAI et al. Learning dexterous in-hand manipulation, *the International Journal of Robotics Research*, 2019.
> > >
> > > [2] Peng et al. Sim-to-real transfer of robotic control with dynamics randomization. *the international conference on robotics and automation (ICRA)*, 2018.
> > >
> > > [3] Hinton et al. Deep neural networks for acoustic modeling in speech recognition: The shared views of four research groups. *IEEE Signal processing magazine*, 2012
> > >
> > > [4] Duchi et al. Adaptive subgradient methods for online learning and stochastic optimization. *Journal of machine learning research*, 2011.

---

> > > > ### Comment · Reviewer_SVo4 · 2021-12-01
> > > > **Updated Rating**
> > > >
> > > > Thanks so much for your renewed response. In light of all the detailed comments, I slightly improved my rating. In summary, I still see this work slightly below the acceptance threshold. My main remaining issue with the work is that it is not yet clearly pointed out what the takeaways for a practitioner would be and in which situations these apply. Experiments on settings slightly different from those covered by the assumption would give an opportunity to discuss whether the theory makes relevant predictions in practically relevant settings where assumptions are not fully satisfied without restricting yourself to toy examples. If there are other works that do such experiments already, then their results should be discussed in light of the newly developed theory.

---

### Official Review · Reviewer_WmdZ · 2021-11-07

**Correctness:** 1
**Technical Novelty And Significance:** 1
**Empirical Novelty And Significance:** Not applicable
**Recommendation:** 8
**Confidence:** 2

**Main Review:**

The paper is well organized and easy to read. Most of the mathematical concepts do come with an intuitive description of their meaning and this allows the reader to get a grasp on the importance of the theoretical results.

The focus of the paper is very relevant to the ML and RL community. Domain randomization is emerging as a fundamental training procedure and theoretical guarantees on its performance are a necessary requirement for application in the real world. The submitted paper not only provides this bounds but also express them in terms of quantities (e.g. the value function gap) which are relevant for downstream real-world applications.

The main limitation of the paper is in assuming no fine-tuning with real-world samples. The concept of "domain randomization oracle" implicitly assumes the ability to eventually find the global optimal policy exploiting history and memory. Somehow this ability shares some commonalities with fine-tuning and therefore it seems possible to extend the author's approach to the more relevant scenario of domain-randomization with fine-tuning.

Another limitation of the paper is related to the format required by the conference. The submission is limited to 9 pages which the authors comply to. However, the proofs of all theorems are in the supplementary material which, as a reviewer, shouldn't be necessary to get most of the paper. This is in this case not really the case because I believe proofs are a necessary component to understand the quality and relevance of the paper.

ADDITIONAL COMMENTS
- Page 5. "The agent does not know explicitly which MDP is sampled, but **she** is allowed to interact with this MDP M for one entire episode.". -> it

- Page 5. "We consider the ideal scenario that the domain randomization algorithm eventually **find** the globally optimal policy of this LMDP,". -> finds



**Summary Of The Paper:**

The paper is a theoretical analysis of domain randomization in the context of latent MDPs. The paper provides bounds on a definition of the sim-to-real gap, defined as the difference between the optimal value function and a "domain randomization oracle" value function. In this context the oracle is defined as a history dependent policy which uses the first steps in the environment to improve its belief on the latent variable and then optimally behaves according to this belief.

Provided bounds can be applied in three different contexts: finite domain randomization (i.e. domain randomization on a finite number of domains) satisfying a separation conditions (i.e. a requirement on the existence of a state-action pair which leads to sufficiently different next states), finite domain randomization without the separation condition and infinite domain randomization.

Bounds are expressed in terms of D (i.e. a bound on number of time steps required to reach any other state in the state space) and of H (i.e. the number of steps in an episode). In the case of finite simulators with separation the domain gap is O(D log^3(H)). In the case of finite simulators without separation the domain gap is O(D H^(1/2) log^(1/2)(H)). In the case of infinite domain randomization the gap is approximatively O(D H^(1/2) log^(1/2)(H)).

Bounds are provided with assumptions and most of these assumptions are justified by proving that removing these assumptions opens the possibility to counterexamples which do not meet the provided bounds.

**Summary Of The Review:**

An important and relevant paper which addresses the problem of computing bounds for the value function of a policy trained with domain randomization and tested on one instance of the latent MDP distribution sampled during training with domain randomization. The paper doesn't have limiting assumptions besides the absence of analyzing how results extend to the fine-tuning case.

As a reviewer, checking the correctness of the paper wasn't easy because all relevant proofs are not in the main paper (9 pages) but mostly in the supplementary material (14 pages). The proof overview (section 6) isn't enough to check the paper correctness.

---

> ### Author Response · Authors · 2021-11-22
> **Response to Reviewer WmdZ**
>
> Thank you for your comments and assessment.
>
> **Regarding the lack of fine-tuning**: We agree typically sim-to-real transfer involve two main steps: the first is to train a policy in the simulator that is transferred to the real-world environment;
> the second is to fine-tune the policy by real-world interactions.
>  In practice, the domain randomization algorithms are often used without  fine-tuning techniques (e.g. [1,2]). The fine-tuning step also involves online RL in the real world (e.g., the policy improvement techniques) which is beyond purely generalizing the knowledge learned from simulation to the real world.
> Our paper focuses on the first step. We establish a theoretical framework, and prove the sim-to-real gap of the policy trained with domain randomization. This policy can be further improved by any fine-tuning techniques with some real-world samples, which is left as an interesting future work.
>
> **Regarding the lack of proofs in the main context**: The space restriction doesn't allow us to include proofs in the main text. We therefore follow the standard approach of theory papers, presenting the main results and explaining key proof techniques in the main text, and providing full proofs in the supplementary materials.
>
> [1] Fereshteh Sadeghi and Sergey Levine. Cad2rl: Real single-image flight without a single real image. *Robotics: Science and Systems*, 2017
>
> [2] OpenAI et al. Learning dexterous in-hand manipulation, *the International Journal of Robotics Research*, 2019.

---

### Decision · Program_Chairs · 2022-01-20

**Decision:**

Accept (Spotlight)

**Comment:**

This manuscript introduces a theoretical framework to analyze the sim2real transfer gap of policies learned via domain randomization algorithms. This work focusses on understanding the success of existing domain randomization algorithms through providing a theoretical analysis. The theoretical sim2real gap analysis requires two critical components: *uniform sampling* and *use of memory*

**Strengths**
All reviewers agree that this manuscript provides a strong theoretical analysis for an important problem (understanding sim2real gap)
well written manuscript, and well motivated
Intuitive understanding for theoretical analysis is provided


**Weaknesses**
analysis is limited to sim2real transfer without fine-tuning in the real world
the manuscript doesn't provide a novel experimental evaluation
lack of take-aways

**Rebuttal**
The authors acknowledge the limitation of not addressing fine-tuning, but also point out that several papers have performed sim2real transfer without fine-tuning.
The authors address the lack of novel experimental evaluation by arguing that the theoretical analysis can be directly linked to existing algorithms for which empirical evaluations have already been performed. I agree with the authors that in that context it seems of little value to redo those experiments. However, I also believe that those links could be made even clearer in the manuscript and I would encourage the authors to do so. Furthermore, while the authors do provide intuitive take-aways for domain randomization algorithms, it would be helpful if those take-aways were more clearly linked to existing algorithms as well (given that there is no experimental evaluation of this).

**Summary**
This manuscript provides a theoretical framework for analyzing the sim2real gap and using that framework provides bounds on the sim2real gap. All reviewers agree this is a strong theoretical analysis. Some take-aways on what makes domain randomization algorithms successful are provided by the provided sim2real-gap analysis (memory use, uniform sampling). Thus I recommend accept.